# Laminin-α2 chain deficiency in skeletal muscle causes dysregulation of multiple cellular mechanisms

Susana G Martins[1,2], Vanessa Ribeiro[1,2], Catarina Melo[1,2], Cláudia Paulino-Cavaco[1,2], Dario Antonini[3],
Sharadha Dayalan Naidu[4], Fernanda Murtinheira[5,6], Inês Fonseca[1,2], Bérénice Saget[1,2], Mafalda Pita[1,2],
Diogo R Fernandes[1,2], Pedro Gameiro dos Santos[1,2], Gabriela Rodrigues[1,2], Rita Zilhão[1,7], Federico Herrera[5,6],
Albena T Dinkova-Kostova[4], Ana Rita Carlos[1,2,*], Sólveig Thorsteinsdóttir[1,2,*]

**LAMA2, coding for the laminin-α2 chain, is a crucial ECM component, particularly abundant in skeletal muscle. Mutations in *LAMA2* trigger the often-lethal *LAMA2*-congenital muscular dystrophy (LAMA2-CMD). Various phenotypes have been linked to LAMA2-CMD; nevertheless, the precise mechanisms that malfunction during disease onset in utero remain unknown. We generated *Lama2*-deficient C2C12 cells and found that *Lama2*-deficient myoblasts display proliferation, differentiation, and fusion defects, DNA damage, oxidative stress, and mitochondrial dysfunction. Moreover, fetal myoblasts isolated from the *dy^W* mouse model of LAMA2-CMD display impaired differentiation and fusion in vitro. We also showed that disease onset during fetal development is characterized by a significant down-regulation of gene expression in muscle fibers, causing pronounced effects on cytoskeletal organization, muscle differentiation, and altered DNA repair and oxidative stress responses. Together, our findings provide unique insights into the critical importance of the laminin-α2 chain for muscle differentiation and muscle cell homeostasis.**

## Introduction

Muscle formation and growth are processes that start early in embryonic development and continue postnatally. In the first wave, during embryonic (primary) myogenesis in the trunk and limbs, muscle stem cells (MuSCs) dissociate from the dermomyotome and migrate to their target sites, marking the areas where the future muscle will develop (Deries et al, 2020). These MuSCs (PAX3+/PAX7+) initiate their differentiation program, which is marked by the activation of the myogenic regulatory factors (MRFs) MYF5, MYOD, MRF4, and myogenin. They then give rise to embryonic myoblasts and finally fuse to form the embryonic (or primary) myotubes, which set the muscle pattern (Deries et al, 2020; Rodriguez-Outeiriño et al, 2021). Although some MuSCs differentiate, others remain stem cells (PAX7+) and proliferate substantially until they receive a second stimulus to start differentiating. The MRFs are again activated, and differentiation and fusion of fetal myoblasts lead to the formation of fetal (secondary) myotubes, marking the beginning of fetal (or secondary) myogenesis. The transition between embryonic and fetal myogenesis is promoted by the transcription factor nuclear factor one X (NFIX). NFIX activates fetal-specific muscle genes and blocks genes characteristic of embryonic myogenesis (Messina et al, 2010; Ribeiro et al, 2023). Myogenesis is therefore a tightly regulated process involving the contribution of distinct players including transcription factors, growth factor–induced signaling pathways, and ECM (Thorsteinsdóttir et al, 2011; Csapo et al, 2020; Zhang et al, 2021).

The ECM is a dynamic and vital noncellular structure within tissues that provides continuous support by forming a scaffold that facilitates the acquisition and maintenance of tissue morphogenesis and homeostasis. The ECM consists of a network of glycoproteins (e.g., laminins, fibronectins, collagens), proteoglycans, and other factors, and its composition and function differ depending on the tissue and the stage of development (Bonnans et al, 2014; Naba et al, 2016). In fact, ECM remodeling is part of normal tissue development and is important for homeostasis and regeneration. During embryonic development, the ECM is constantly changing as tissues and organs are formed and new functions are acquired (Bonnans et al, 2014). The ECM, especially the basement membrane (i.e., a sheet-like specialized type of ECM), plays a crucial role in ensuring a proper communication between the extracellular and intracellular environments, as well as the

[1]Centre for Ecology, Evolution and Environmental Changes (CE3C) & CHANGE, Faculdade de Ciências, Universidade de Lisboa, Lisboa, Portugal [2]Departamento de Biologia Animal, Faculdade de Ciências, Universidade de Lisboa, Lisboa, Portugal [3]Department of Biology, University of Naples "Federico II", Naples, Italy [4]Jacqui Wood Cancer Centre, Division of Cellular and Systems Medicine, School of Medicine, University of Dundee, Dundee, UK [5]Biosystems and Integrative Sciences Institute (BioISI), Faculdade de Ciências, Universidade de Lisboa, Lisboa, Portugal [6]Departamento de Química e Bioquímica, Faculdade de Ciências, Universidade de Lisboa, Lisboa, Portugal [7]Departamento de Biologia Vegetal, Faculdade de Ciências, Universidade de Lisboa, Lisboa, Portugal

Correspondence: arcarlos@ciencias.ulisboa.pt; solveig@ciencias.ulisboa.pt
*Ana Rita Carlos and Sólveig Thorsteinsdóttir are cosenior authors

activation of signaling cascades important for cell function (Bonnans et al, 2014; Naba et al, 2016). Laminins are a family of conserved trimeric ECM proteins that are core components of basement membranes (Domogatskaya et al, 2012). Laminin isoforms are named according to their chain composition, where a laminin formed by α1, β1, and γ1 chains is designated laminin 111 (Aumailley et al, 2005).

In skeletal muscle, the ECM plays a critical role in regulating muscle development, growth, and repair, and is vital for effective muscle contraction and force transmission (Gillies & Lieber, 2011; Csapo et al, 2020; Deries et al, 2020). Not surprisingly, mutations in genes encoding ECM components have been linked to a group of conditions collectively termed muscular dystrophies. These conditions are characterized by progressive muscle weakness and the loss of muscle mass. One such condition is *LAMA2*-congenital muscular dystrophy (LAMA2-CMD), which is caused by mutations in the *LAMA2* gene, codifying for the α2 chain of laminin 211 (the most abundant laminin isoform in the basement membrane of adult skeletal muscle) and laminin 221 (Yurchenco et al, 2018; Gawlik & Durbeej, 2020). LAMA2-CMD is a devastating and often fatal neuromuscular disease, in which patients display hypotonia from birth. Studies using postnatal mouse models of the disease and biopsies from LAMA2-CMD patients have identified several mechanisms linked to the pathology of the disease, including proliferation and differentiation defects, along with increased oxidative stress and metabolic alterations (de Oliveira et al, 2014; Fontes-Oliveira et al, 2017; Yurchenco et al, 2018; Gawlik et al, 2019; Kölbel et al, 2019; Gawlik & Durbeej, 2020; Harandi et al, 2020; Martins et al, 2021). In addition, patients suffering from this pathology develop chronic inflammation and fibrosis, which are considered to be secondary events that arise from constant ECM remodeling and cycles of degeneration and regeneration of the muscle fibers (Yurchenco et al, 2018; Gawlik & Durbeej, 2020). To date, most of the studies aimed at unraveling the mechanisms underlying this disease have been performed postnatally, after the emergence of secondary events. However, our previous data using the $dy^W$ mouse model for LAMA2-CMD indicate that disease onset occurs during fetal development, specifically between embryonic days (E) 17.5 and 18.5 (Nunes et al, 2017). This onset is characterized by a reduction in the number of MuSCs/myoblasts and impaired muscle growth, despite the muscle appearing morphologically normal (Nunes et al, 2017). In the present study, we designed both in vitro and in vivo approaches to dissect the molecular and cellular processes underlying the onset of LAMA2-CMD. We reasoned that these processes could represent the core mechanisms that are disrupted in the absence of a functional laminin-α2 chain, thus being triggers of the disease. Our results revealed that the absence of the laminin-α2 chain impairs myoblast proliferation, differentiation, and fusion, and triggers DNA damage and oxidative stress. RNA-sequencing analysis revealed a massive down-regulation of gene expression in muscle fibers at E17.5, indicating compromised muscle differentiation and alterations in the cytoskeleton. Our findings suggest that the absence of *Lama2* disrupts the extracellular-to-intracellular communication, as well as cell biomechanics, compromising muscle differentiation and muscle fiber structure and function. The identification of these profound perturbations already at fetal stages, when the muscle appears morphologically normal and

before secondary events occur, indicates that they are part of the primary disease mechanism that triggers the disease.

# Results

## *Lama2* deficiency leads to proliferation and cell cycle defects in vitro

Most research on LAMA2-CMD has been performed using in vivo models (Yurchenco et al, 2018; Gawlik & Durbeej, 2020). Although this is of critical importance, the severity and complexity of the disease present challenges in precisely identifying the mechanisms triggering the onset of LAMA2-CMD. This is particularly relevant given that the vast majority of previous studies used models where the secondary events of the disease were already present, potentially obscuring the primary events. To tackle this issue, we generated an in vitro model for LAMA2-CMD using the myoblast cell line C2C12 (Fig 1A), which has been extensively used as an in vitro model to study myogenesis (Burattini et al, 2004; Biressi et al, 2007; Messina et al, 2010; Rossi et al, 2016). *Lama2* was deleted using the CRISPR/Cas9 technology, and deletion was accessed by quantitative PCR (qPCR) for each established single-cell clone, as represented in Fig 1B. At least two distinct single-cell clones were used for each experiment to ensure that the observed phenotype was due to the *Lama2* deletion and not to potential off-target or clonal effects. The characterization of the mutations was performed by Sanger sequencing of the PCR products covering the gRNA/Cas9 target regions in *Lama2* exons 4 and 9 (Fig S1A–C).

Our previous findings, using the $dy^W$ mouse model for LAMA2-CMD, showed that the disease onset is marked by a significant reduction in the number of MuSCs and myoblasts (Nunes et al, 2017). To test whether *Lama2* deficiency affects cell proliferation, we compared *Lama2* knockout and WT C2C12 cells and observed a significant decrease in proliferation in the *Lama2* knockout cells (Fig 1C). This proliferation defect may be explained by an arrest of *Lama2* knockout cells in the G1 phase of the cell cycle and a concomitant reduction in the S phase (Figs 1D and S1D). Because of this defect, *Lama2* knockout single-cell clones were passaged for a limited time and new clones were periodically generated and validated as described above.

## Absence of *Lama2* triggers oxidative stress and DNA damage

Cell cycle arrest can be caused by various insults, such as oxidative stress and DNA damage (Stallaert et al, 2022). Given the significant role of oxidative stress and mitochondrial dysfunction in LAMA2-CMD pathology (Millay et al, 2008; de Oliveira et al, 2014; Fontes-Oliveira et al, 2017; Kölbel et al, 2019; Harandi et al, 2020), we analyzed different markers for these insults in our in vitro model (Fig 1E–L). To evaluate the overall redox status of *Lama2*-deficient cells, we measured the levels of reduced glutathione (GSH), an important endogenous antioxidant. We observed a significant decrease in GSH in *Lama2* knockout cells in comparison with their WT counterparts (Fig 1E), suggesting an increase in the oxidative status of *Lama2* knockout cells. The decrease in GSH is not likely to be associated

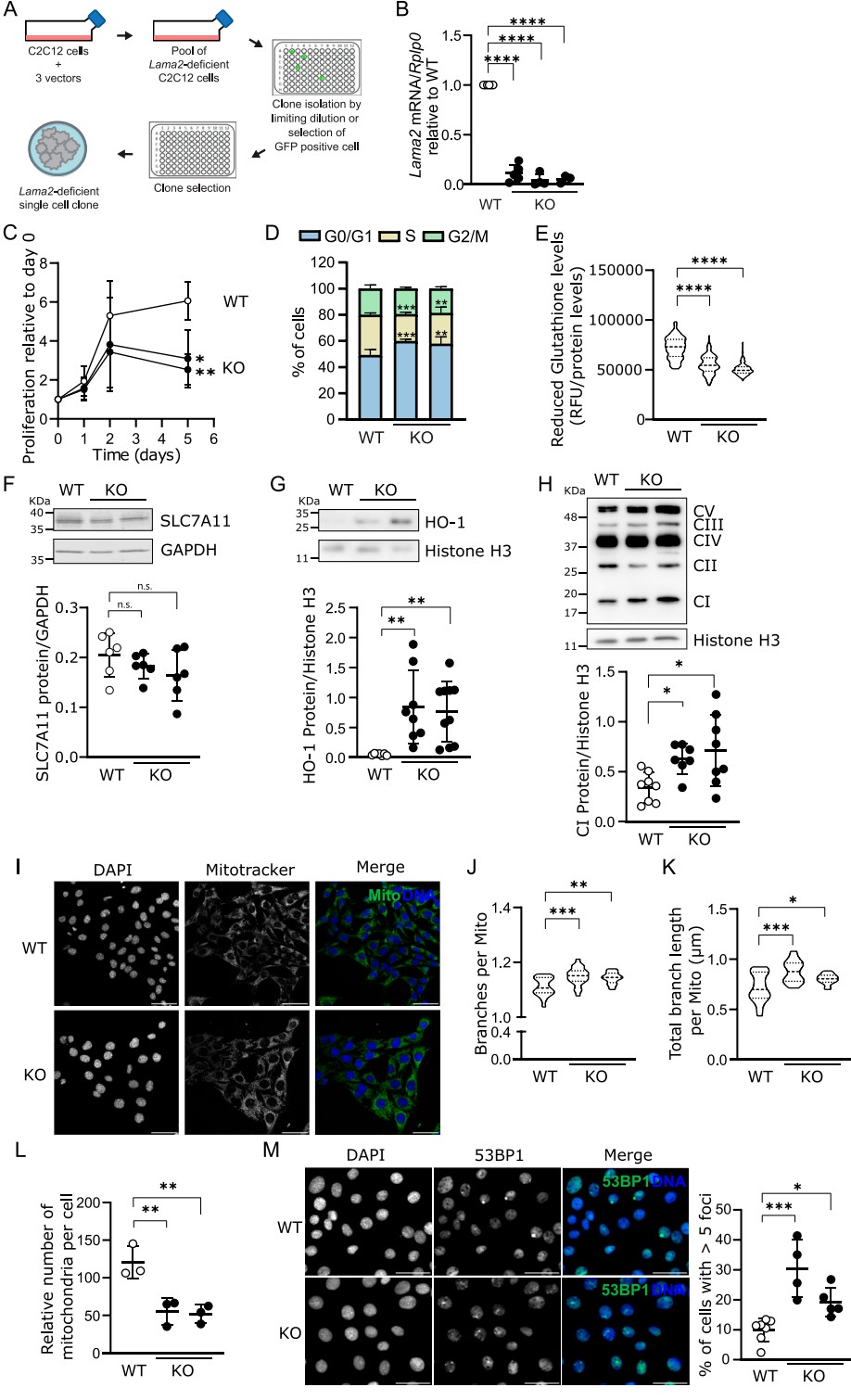

**Figure 1. *Lama2* deficiency impairs proliferation, possibly because of cell cycle arrest, and triggers DNA damage and oxidative stress.**
**(A)** Schematic representation of *Lama2* knockout cell line generation using the CRISPR/Cas9 technology. **(B)** RNA was extracted from C2C12 WT and three independent *Lama2* knockout single-cell clones (KO), and the expression of *Lama2* was analyzed by qPCR. *Lama2* expression levels were normalized to the housekeeping gene *Rplp0* and then to WT. n = 3–7 independent samples per group were collected independently. **(C)** WT and KO C2C12 cells were plated, and cell proliferation was monitored on days 0, 1, 2, and 5 using resazurin. Data were plotted relative to day 0, in order to analyze the proliferation rate. n = 3 independent experiments, each with four technical replicates. **(D)** WT and KO C2C12 cells were stained with propidium iodide (PI) and analyzed by flow cytometry. The percentage of cells in the G1/G0, S, and G2/M phase of the cell cycle is represented. n = 3–5 experiments per cell line. **(E)** To measure the levels of reduced glutathione, WT and KO C2C12 cells were incubated with 40 *μ*M of monochlorobimane for 30 min, and fluorescence was measured (excitation: 390 nm; emission: 490 nm). Fluorescence levels were normalized for protein levels (BCA quantification method). n = 3–4 independent experiments, each with 36 independent measurements. **(F, G, H)** WT and KO C2C12 cells were harvested for Western blot analysis with anti-SLC7A11 (F), anti-HO-1 (G), or anti-oxidative phosphorylation (CI, complex I; CII, complex II; CIII, complex III; CIV, complex IV; CV, complex V) (H) antibodies. Histone H3 or GAPDH was used as loading controls. Two KO lanes represent experiments performed with two independent *Lama2* knockout single-cell clones. n = 6–10 samples per group were collected independently. Densitometry analysis is shown under each Western blot. **(I)** Representative images of WT and KO C2C12 cells incubated with the MitoTracker mitochondrial dye, fixed, and then processed for fluorescence microscopy. DAPI was used to counterstain nuclei. n = 3 independent experiments. Scale bar: 50 *μ*m. **(I, J, K)** Quantification of the number of branches per mitochondria (J) and the total branch length per mitochondrion (K) in cells treated as in (I). n = 21 images per genotype, analyzed from three independent experiments (seven images per experiment). **(L)** Total DNA was extracted from WT and C2C12 cells and analyzed by qPCR. The ratio between *Nd1* DNA expression (encoded in the mitochondria) and *Hk2* DNA expression (encoded in the nucleus) was calculated and used as a proxy for the mitochondrial number per cell. n = 3 independent experiments. **(M)** Representative images of WT and KO C2C12 cells fixed and processed for immunofluorescence with an anti-53BP1 antibody, a DNA damage marker. DAPI was used to counterstain nuclei. n = 4–7 independent experiments. Scale bar: 50 *μ*m. Quantification of the number of cells with more than five 53BP1 foci was performed. Images of two *Lama2* knockout single-cell clones were analyzed separately. **(B, C, D, E, F, G, H, J, K, L, M)** Statistical analysis was performed using ordinary one-way ANOVA with Dunnett's multiple comparisons test for (B, E, F, G, H, J, K, L, M), and two-way ANOVA with Tukey's multiple comparisons test for (C, D). Data are represented as the mean ± SD. *P*-value: *P < 0.05, **P < 0.01, ***P < 0.001, ****P < 0.0001. The ROUT or Grubbs methods were used to identify outliers.

with defective glutathione synthesis, as indicated by the normal levels of GCLC and GCLM (Fig S2A), the two subunits of the enzyme catalyzing the rate-limiting step in the biosynthesis of GSH. However, we found that the cystine/glutamate transporter SLC7A11 showed a trend toward reduction in *Lama2*-deficient cells (Fig 1F). This potentially indicates an impairment in the transport of cystine, and consequently in the availability of cysteine, a precursor for glutathione synthesis. In accordance with the lower levels of GSH, we also found that the stress-induced heme oxygenase 1 (HO-1) was up-regulated in *Lama2* knockout cells in comparison with their WT counterparts (Fig 1G).

The mitochondrial electron transport chain is an important source of reactive oxygen species (ROS), and mitochondrial dysfunction is therefore closely linked to oxidative stress (Bhatti et al, 2017). To investigate whether the mitochondrial function was altered in vitro, we analyzed the levels of the oxidative phosphorylation (OXPHOS) complexes (Figs 1H and S2B). Complex I was increased in *Lama2*-deficient cells (Fig 1H), whereas the remaining complexes remained largely unchanged (Fig S2B). Complex I is a key source of ROS production in the mitochondria, and increased levels of complex I have been associated with aging (Miwa et al, 2014; Signorile et al, 2023). Notably, an increase in mitochondrial branching (Leduc-Gaudet et al, 2015) and a decrease in mitochondrial number (Crane et al, 2010) have also been associated with skeletal muscle aging. Therefore, we next analyzed the mitochondrial branching (Fig 1I–K) and the mitochondrial number (Fig 1L) using *Lama2*-proficient versus *Lama2*-deficient cells. We found a significant increase in both the number of mitochondrial branches and their total length in *Lama2*-deficient cells when compared to WT cells (Fig 1J and K). This suggests that in the absence of *Lama2*, mitochondria decrease their fission and increase their fusion levels, as previously shown to occur in aged muscle (Leduc-Gaudet et al, 2015). In keeping with this, we also found a significant reduction in the number of mitochondria in *Lama2*-deficient cells (Fig 1L). These data give further support to the notion that *Lama2*-deficient cells display increased ROS, which compromises mitochondrial integrity and may impair cell proliferation.

To further tackle the potential causes of impaired proliferation and cell cycle arrest, we investigated whether the absence of *Lama2* could also lead to DNA damage, which can be triggered by various insults including increased ROS levels (Martins et al, 2021). To evaluate DNA damage, we analyzed the percentage of cells displaying more than five 53BP1 foci and found that this percentage was significantly increased in *Lama2*-deficient cells when compared to their WT counterparts (Fig 1M). Hence, a significant increase in DNA damage may also be involved in the early stages of *Lama2* deficiency.

### *Lama2* deficiency disrupts normal cell fate mechanisms

Considering that our data suggest that *Lama2* deficiency may lead to proliferation defects, increased oxidative stress, and increased DNA damage in C2C12 cells (Fig 1), we went on to test whether these changes could lead to the induction of apoptosis (Fig S2C). A comparison of the mRNA expression of the pro-apoptotic *Bax* gene and the anti-apoptotic *Bcl2* gene between *Lama2*-deficient and *Lama2*-proficient C2C12 cells did not show significant differences

(Fig S2C). Previous studies have demonstrated the importance of autophagy in preventing MuSC senescence and apoptosis (White et al, 2018; Chang, 2020) and in supporting proper myogenesis and muscle differentiation (Fortini et al, 2016). We therefore tested whether *Lama2* deficiency could lead to impaired autophagy. We analyzed the levels of the autophagosome marker LC3 and found that LC3-II was significantly decreased in *Lama2*-deficient C2C12 cells in comparison with the WT (Fig S2D). This reduction was also apparent for LC3-I, but because it was often undetectable in *Lama2*-deficient cells, quantification was not possible. In addition, we analyzed the expression of two essential components of the conventional autophagy pathway, ATG5 and ATG7 (Arakawa et al, 2017), and found that *Atg7* was down-regulated in *Lama2*-deficient C2C12 cells when compared to their WT counterparts (Fig S2E). Together, these findings indicate defective autophagosome formation in *Lama2*-deficient cells.

### Myotube formation is impaired in *Lama2*-deficient cells

Because we observed a decrease in the proliferation rate, we decided to analyze myoblast differentiation using this model. Typically, a decrease in the proliferation rate or cell cycle arrest is associated with a change in the differentiation program, given the sequential nature of these processes (Deries & Thorsteinsdóttir, 2016). Moreover, if defective myoblast differentiation is observed, it could compromise the formation, growth, and/or the stability of muscle fibers. To analyze whether differentiation is impaired in the absence of *Lama2*, we induced differentiation in *Lama2*-deficient and *Lama2*-proficient C2C12 cells in vitro and followed the formation of myotubes for 14 d (Fig S3A). To avoid the confounding effect of the delayed proliferation observed in the *Lama2*-deficient cells (Fig 1C and D), the differentiation experiments were performed using normal proliferation medium (containing 10% FBS) and cells were allowed to differentiate by the cell–cell contact. In addition, cells were plated at high density to allow confluency to be reached within the first 48 h (Fig S3A), before significant differences in proliferation. *Lama2*-proficient cells formed evident myotube-like structures by day 5, whereas similar structures, even though rare, could only be observed from day 12 onward in *Lama2*-deficient cells (Fig S3A). To determine whether the aligned cellular structures observed under the brightfield microscope were indeed myotubes, we stained *Lama2*-deficient and *Lama2*-proficient cells with an anti-myosin heavy chain (MyHC) antibody, a protein synthesized by myotubes, but not by myoblasts (Fig 2A and B). MyHC was detected in WT cells already at day 5 post-differentiation (Fig 2A), confirming the formation of myotubes, defined as cells with more than two nuclei. In contrast, *Lama2*-deficient cells presented rare and insipient myotube formation with only one or two nuclei at day 5 (Fig 2A and A'). Despite the increase in the number of aligned myoblasts by day 14, the number of fibers per total number of nuclei was still significantly reduced in the *Lama2*-deficient cells, and the vast majority of MyHC-positive cells had only one nucleus (Fig 2B and B'). To further extend this analysis, we isolated primary fetal mouse myoblasts from WT and $dy^W$ fetuses and found that $dy^W$ myoblasts display a differentiation defect, which is evident by the significant reduction in the number of myofibers formed when compared to WT, already 3 d after being placed in differentiation

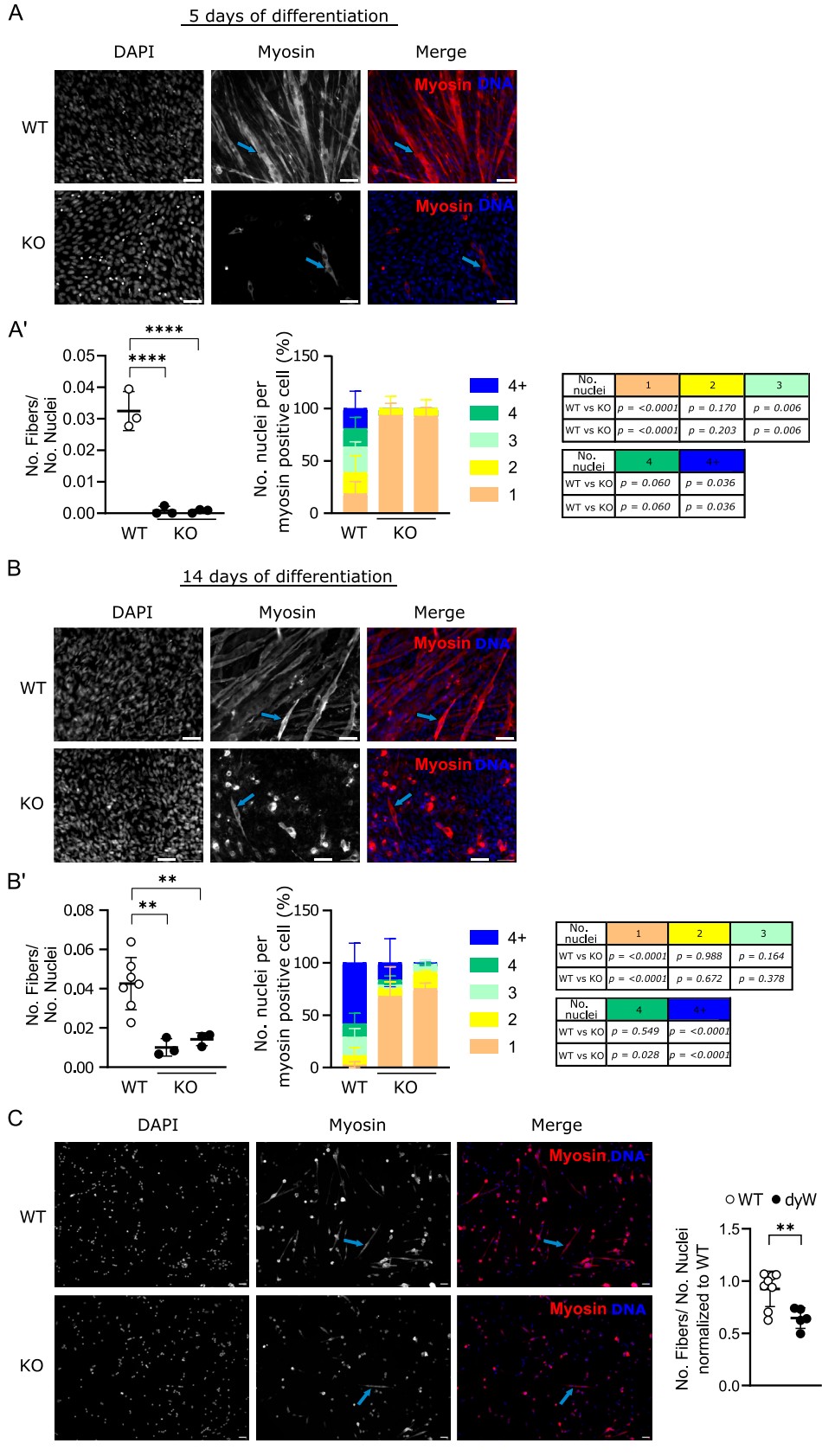

**Figure 2.  *Lama2* knockout cells show impaired differentiation.**

**(A)** C2C12 WT and two independent *Lama2* knockout single-cell clones (KO) were plated (day −1), cultured until day 5 to induce differentiation, and then fixed and processed for immunofluorescence with an anti-myosin heavy chain antibody. Representative images are shown. DAPI was used to counterstain nuclei. n = 3 independent experiments. Scale bar: 50 μm. Blue arrows indicate myosin-positive cells. **(A′)** Ratio between the number (No.) of fibers (myosin-positive cells with two or more nuclei) and the total number of nuclei was quantified (left panel), as well as the number of nuclei per myosin-positive cell (middle panel) and the respective *P*-values (right panel). **(B)** C2C12 WT and two independent *Lama2* single-cell clones (KO) were plated (day −1) and cultured until day 14 to induce differentiation, and then fixed and processed for immunofluorescence with an anti-myosin heavy chain antibody. Representative images are shown. Counterstaining of nuclei was performed with DAPI. n = 3–7 independent experiments. Scale bar: 50 μm. Blue arrows indicate myosin-positive cells. **(B′)** Ratio between the number (No.) of fibers (myosin-positive cells with two or more nuclei) and the total number of nuclei was quantified (left panel), as well as the number of nuclei per myosin-positive cell (middle panel) and respective *P*-values (right panel). **(C)** Primary mouse myoblasts were isolated from WT and *dyW* E16.5–E19.5 fetuses, left to adhere and proliferate for 4 d, and then incubated in differentiation medium for an additional 3 d. Cells were then fixed and processed for immunofluorescence with an anti-myosin heavy chain antibody. Representative images are shown on the left. DAPI was used to counterstain nuclei. n = 5–8 fetuses per genotype. Scale bar: 50 μm. Blue arrows indicate myosin-positive cells. Quantification of the number of myofibers (i.e., myosin-expressing cells that contain two or more nuclei) per total number of nuclei is shown on the right. **(A′, B′, C)** Statistical analysis in (A′, B′, C) was performed using ordinary one-way ANOVA with Dunnett's multiple comparisons test. Data are represented as the mean ± SD. *P*-value: *P < 0.05, **P < 0.01, ***P < 0.001, ****P < 0.0001. The Grubbs method was used to identify outliers.

medium (Fig 2C). This indicates a severe differentiation defect in the absence of *Lama2*.

## Myoblast differentiation and fusion are impaired in *Lama2*-deficient cells

To better understand which mechanisms were involved in this differentiation defect, we analyzed the expression of several myogenic markers in C2C12 cells grown under proliferation conditions (Fig 3). The mRNA expression of the MRF myogenin (*Myog*) (Fig 3A) was significantly lower in *Lama2*-deficient cells in comparison with their WT counterparts. Moreover, both the number of MYF5-positive nuclei (Fig 3B) and MYF5 nuclear protein levels (Fig 3C) were significantly reduced in *Lama2*-deficient cells compared with WT cells. This decrease was concomitant with an increase in NFIX nuclear protein levels (Fig 3C), and in the number of NFIX-positive nuclei (Fig 3D). NFIX is a transcription factor responsible for the transition between embryonic and fetal myogenesis (Messina et al, 2010; Ribeiro et al, 2023), and it has been demonstrated to be highly expressed in C2C12 cells (Biressi et al, 2007). These results suggest that *Lama2* deficiency leads to increased NFIX levels, possibly causing a shift toward aberrant fetal myogenesis, compromising the skeletal muscle differentiation process. To determine whether later stages of myogenesis were affected by *Lama2* deficiency, we analyzed the mRNA expression of the tubulin b6 chain (*Tubb6*) and myosin light chain (*Myl1*), both myogenin target genes, at day 5 post-differentiation (Fig 3E and F). Both genes, previously shown to be important players in differentiation (Burguière et al, 2011; Park et al, 2018; Randazzo et al, 2019), were significantly reduced in *Lama2*-deficient cells when compared to WT cells (Fig 3E and F).

Because we have shown that autophagy is compromised in *Lama2*-deficient C2C12 myoblasts (Fig S2D and E) and that autophagy is known to be important for myoblast differentiation (McMillan & Quadrilatero, 2014), we investigated whether the expression of *Atg5* and *Atg7* was impaired in *Lama2*-deficient cells at day 5 post-differentiation (Fig 3G). We found that the expression of both *Atg5* and *Atg7* is significantly down-regulated in *Lama2*-deficient cells in comparison with the WT.

To understand whether the differentiation defect could also be linked with an impairment in myoblast fusion, we analyzed the expression of two key players in myoblast fusion, Myomaker and Myomerger (also known as Myomixer or Minion), which act in a sequential manner to promote plasma membrane remodeling and fusion (Leikina et al, 2018). Myomaker acts first, promoting hemifusion of the adjacent plasma membranes, whereas Myomerger completes the fusion process (Leikina et al, 2018). Our results showed that *Lama2*-deficient cells have impaired *Mymk* expression (encoding Myomaker) (Fig 3H), indicating that fusion is compromised already at the hemifusion stage. In addition, there is a trend toward the reduced expression of *Mymx* (encoding Myomerger) (Fig 3I). This suggests that *Lama2*-deficient C2C12 myoblasts lack the ability to fuse.

Overall, these results indicate that *Lama2* deficiency impairs the essential cytoskeletal and plasma membrane organization required for the transition from myoblasts to myotubes.

## The absence of *Lama2* in fetal muscle fibers leads to a significant down-regulation in gene expression

Our in vitro findings with the C2C12 LAMA2-CMD cellular model highlight the importance of *Lama2* in myoblast differentiation and myotube formation (Figs 2, 3, and S2). Indeed, myotube formation is also impaired in primary fetal myoblasts isolated from $dy^W$ fetuses and differentiated in vitro (Fig 2C). Nevertheless, a normal number of muscle fibers form during in vivo myogenesis in $dy^W$ fetuses, but they fail to grow as much as in WT fetuses (Nunes et al, 2017). Therefore, to determine the effect of *Lama2* deficiency on fetal muscle fibers, we performed an RNA-sequencing (RNAseq) analysis on muscle fibers isolated from E17.5 fetuses, the developmental stage at which the first signs of the disease have been reported (Nunes et al, 2017). The RNAseq analysis of muscle fibers revealed a profound down-regulation of gene expression with 4,958 genes down-regulated in $dy^W$ fetuses, when compared to the WT (adjusted *P*-value of 0.01, $\log_2$ fold change of ±1.5), whereas only 181 genes were up-regulated (Fig 4A). This indicates that *Lama2* deficiency has a severe impact on the gene expression program of muscle fibers, potentially affecting their stability. To gain further insights into the pathways altered in the absence of *Lama2*, we conducted a functional analysis. The result revealed that down-regulated genes were significantly associated with pathways related to plasma membrane structure and function, transmembrane signaling, intermediate filament organization, and mechanisms involving the maintenance of the cellular redox status (Fig 4B). This supports the notion that the absence of the α2 chain of laminin causes substantial structural changes in the link between the basement membrane and the cell cytoskeleton. On the contrary, up-regulated genes were involved in protein and RNA processing and metabolism and in collagen-containing ECM (Fig 4C). This increase in collagen is in accordance with previous findings in postnatal/adult muscle showing that fibrosis is a central hallmark of LAMA2-CMD pathogenesis (Taniguchi et al, 2006; Accorsi et al, 2020).

## *Lama2* is crucial for gene expression regulating muscle fiber differentiation

Given the possible defect in muscle differentiation, we next compared the differentially expressed genes (DEGs) in fetal $dy^W$ muscle fibers versus WT controls, with a previously published ChIP-seq database (10.17989/ENCSR000AHO, ENCODE), where C2C12 myoblasts and C2C12 myocytes (i.e., differentiated cells) were analyzed by histone chromatin immunoprecipitation sequencing (ChIP-seq) for different histone modifications. In particular, we compared the DEGs from our muscle fiber RNAseq with C2C12 myoblast and C2C12 myocyte ChIP-seq analysis for histone H3 trimethylation on lysine 4 (H3K4me3) (Fig 4D), a gene activation marker. 191 genes were activated specifically in C2C12 myocytes (i.e., under differentiation conditions) and down-regulated in $dy^W$ fetal muscle fibers (Fig 4D, Table S1). Functional analysis showed that these genes are critical for cell differentiation, including striated muscle cell differentiation, as well as cytoskeletal structure (Fig 4E). These data strongly support the hypothesis that muscle fiber differentiation is also impaired in $dy^W$ fetuses.

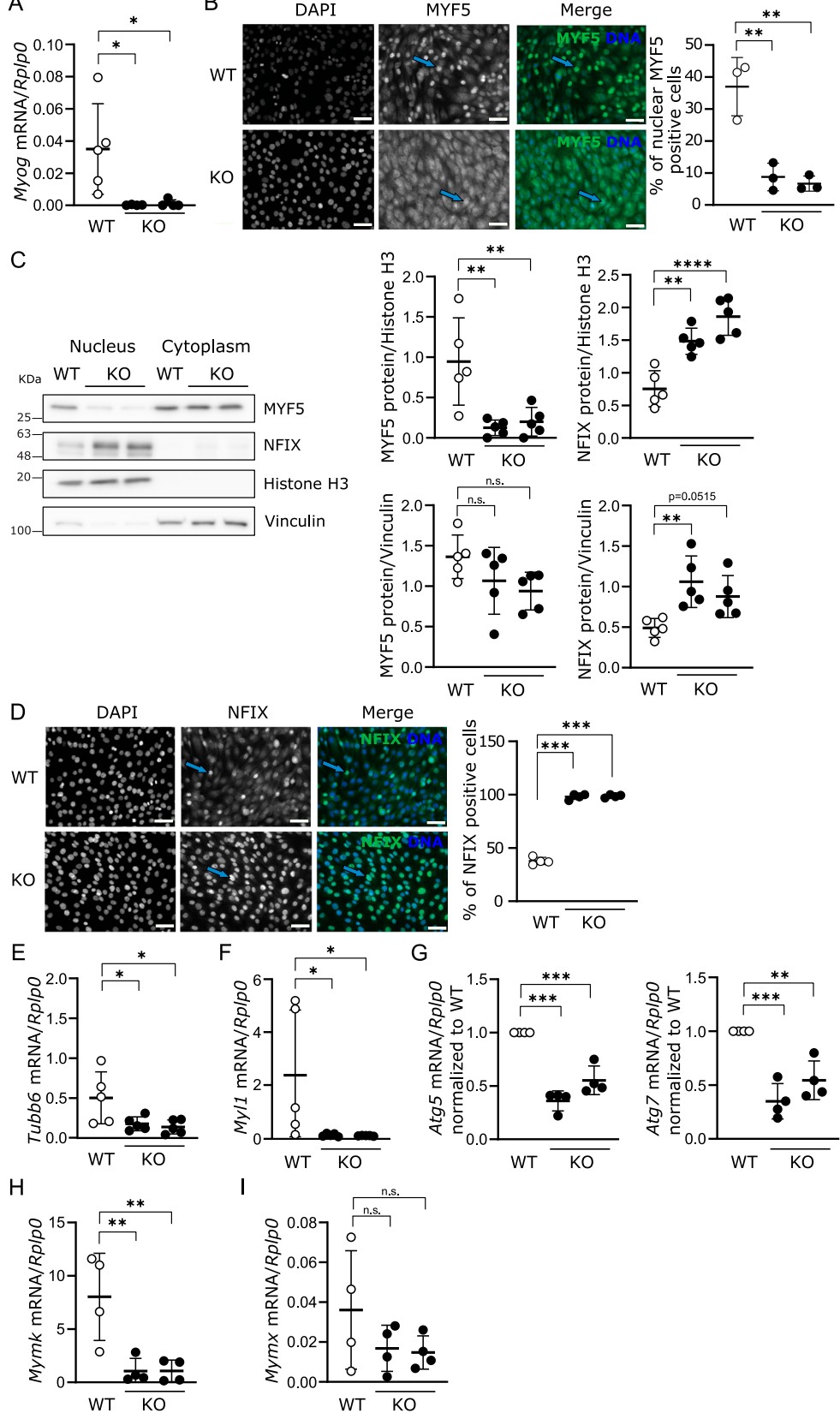

**Figure 3. *Lama2* knockout cells show impaired differentiation and alterations in MYF5 and nuclear factor one X (NFIX) pathways.**
**(A)** RNA was extracted from C2C12 WT and two independent *Lama2* knockout single-cell clones (KO), and the expression of *Myogenin* (*Myog*) was analyzed by qPCR. Expression levels were normalized to the housekeeping gene *Rplp0*. n = 4–5 independent samples per group were collected independently. **(B)** Representative images of WT and KO C2C12 cells fixed and processed for immunofluorescence with an anti-MYF5 antibody are shown. DAPI was used to counterstain nuclei. Images of two *Lama2* knockout single-cell clones were analyzed separately. Quantification of the number of nuclear MYF5-positive cells is shown on the right. n = 3 independent experiments. Blue arrows indicate MYF5 nuclear staining. Scale bar: 50 µm. **(C)** Nuclear and cytoplasmic fractions of WT and C2C12 cells were separated by cell fractionation. Protein was extracted from each fraction and analyzed by Western blot with anti-MYF5 and anti-NFIX antibodies (left panel). Histone H3 (nuclear phase) and vinculin (cytoplasmic phase) were used as loading controls. Two KO lanes represent experiments performed with two independent *Lama2* knockout single-cell clones. n = 5 samples per group were collected independently. Densitometry analysis is shown on the right. **(B, D)** Immunofluorescence analysis similar to the one described in (B) using an anti-NFIX antibody. Images of two *Lama2* knockout single-cell clones were analyzed separately. Quantification of the number of nuclear NFIX-positive cells is shown on the right. n = 3 independent experiments. Blue arrows indicate NFIX nuclear staining. Scale bar: 50 µm. **(E, F)** Myogenin target genes *Tubb6* (E) and *Myl1* (F) were analyzed by qPCR in WT and KO C2C12 cells on day 5 of differentiation. Expression levels were normalized using the housekeeping gene *Rplp0*. n = 5 samples per group were collected independently. **(G)** Expression of the autophagy markers *Atg5* and *Atg7* was analyzed by qPCR in WT and KO C2C12 cells on day 5 of differentiation. Expression levels were normalized using the housekeeping gene *Rplp0* and then normalized to the WT controls. n = 4 samples per group were collected independently. **(H, I)** Expression of myoblast fusion markers *Mymk* (H) and *Mymx* (I) was analyzed by qPCR in WT and KO C2C12 cells, on day 5 of differentiation. Expression levels were normalized using the housekeeping gene *Rplp0*. n = 4 samples per group were collected independently. **(A, B, C, D, E, F, G, H, I)** Statistical analysis in (A, B, C, D, E, F, G, H, I) was performed using ordinary one-way ANOVA with Dunnett's multiple comparisons test. Data are represented as the mean ± SD. *P*-value: **P* < 0.05, ***P* < 0.01, ****P* < 0.001, *****P* < 0.0001.

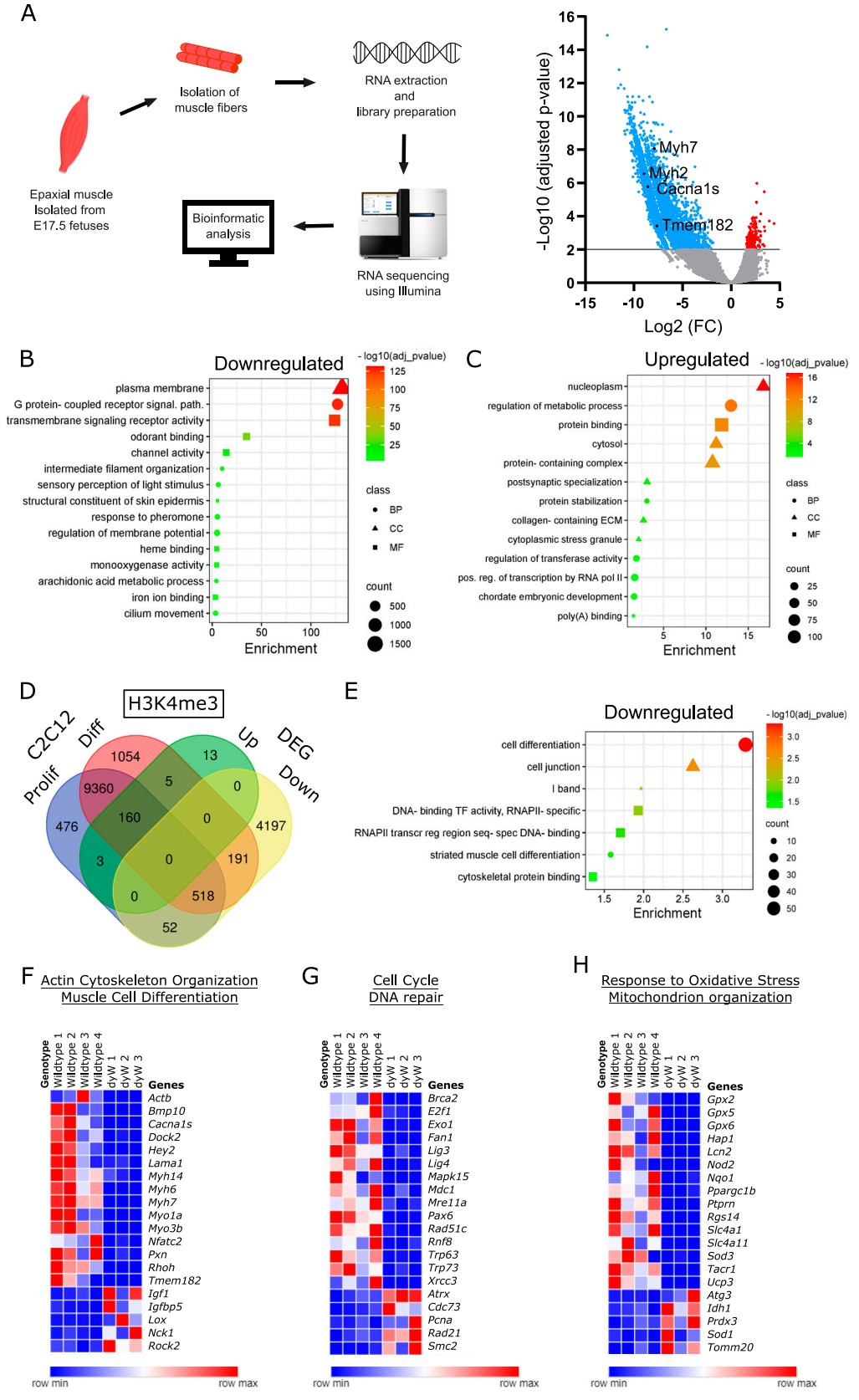

To broaden our analysis of the pathways that were altered in $dy^W$ versus WT fetal muscle fibers, we compared the DEGs of WT versus $dy^W$ muscle fibers with genes from various ontologies with relevance to our findings (Fig 4F–H, Tables S2, S3, S4, S5, S6, and S7). This analysis showed perturbations in actin cytoskeleton organization and muscle cell differentiation, including the down-regulation of *Cacna1s*, *Myh7*, and *Tmem182* (Fig 4F, Tables S2 and S3), some of the top down-regulated genes identified in our RNAseq analysis (Fig 4A). *Cacna1s* and *Tmem182* were also identified as common genes in our comparison between DEGs from our muscle fiber RNAseq with C2C12 myoblast and C2C12 myocyte ChIP-seq analysis for H3K4me3 (Table S1). In addition, we compared DEGs with cell cycle and DNA repair gene ontologies, where genes such as the cell cycle regulator *E2f1* and the DNA repair genes *Exo1* and *Lig4* were found to be down-regulated (Fig 4G, Tables S4 and S5). Comparison of DEGs with genes involved in oxidative stress response and mitochondrion organization revealed the down-regulation of several antioxidant genes such as *Gpx2*, *Gpx5*, *Gpx6*, and *Nqo1* (Fig 4H, Tables S6 and S7). Collectively, these data show that the absence of *Lama2* in fetal muscle fibers has a key impact on multiple pathways.

### *Myh2*, *Myh7*, *Tmem182*, and *Cacna1s* are down-regulated in the absence of *Lama2*

To validate the important defect in fetal muscle fiber differentiation identified in our RNAseq analysis (Fig 4), we analyzed four top hits: the fast MyHC gene *Myh2*, previously shown to be implicated in skeletal muscle differentiation; the slow MyHC gene *Myh7*, typically expressed during embryonic myogenesis and in slow myofibers (Pette & Staron, 2000; Schiaffino et al, 2015; Chatterjee et al, 2019; Agarwal et al, 2020); the MYOD target gene *Tmem182*, a negative regulator of muscle growth (Luo et al, 2021); and the $\alpha$1 s subunit of the voltage-gated L-type $Ca^{2+}$ channel (Cav1.1) (*Cacna1s*), recently shown to be required for muscle fiber maturation (Dos Santos et al, 2023) (Fig 5). A comparison between the normalized counts of the RNAseq (Fig 5A–D) and qPCR results of isolated muscle fibers from E17.5 WT and $dy^W$ fetuses (Fig 5E–H) confirmed the down-regulation of these genes. This additional evidence supports the presence of a defective muscle differentiation signature in $dy^W$ fetuses.

We also analyzed whether the expression of these genes was impaired in our in vitro model for LAMA2-CMD under differentiation conditions. After 5 d of differentiation, *Tmem182* and *Cacna1s* were

significantly down-regulated in *Lama2*-deficient cells (Fig 5I and J), whereas no significant difference was observed in *Myh7* expression, despite a tendency to be reduced in the absence of *Lama2* (Fig S3B). *Myh2* expression did not show differences under our experimental conditions (Fig S3B). However, *Myh7* was significantly decreased after 14 d of differentiation (Fig 5K), consistent with the defective differentiation phenotype found in vitro (Figs 2, 3, and S3). The expression of both *Tmem182* and *Cacna1s* was also found to be reduced in $dy^W$ primary fetal myoblasts, when compared to WT myoblasts, 3 d after the addition of differentiation medium (Fig 5L and M).

Given that *Cacna1s* expression was found to be reduced both in vivo and in vitro, we quantified intracellular calcium levels in our in vitro model. Intracellular calcium levels were significantly higher in *Lama2*-deficient cells, compared with their WT counterparts (Fig 5N), possibly hinting at a deregulation in calcium import and export.

TMEM182 has been shown to negatively regulate muscle growth by interacting directly with integrin $\beta$1 (ITGB), reducing both its binding to laminin and the ITGB1-dependent FAK activity (Luo et al, 2021). We therefore analyzed FAK phosphorylation (P-FAK) in whole muscles of E17.5 WT and $dy^W$ fetuses. We found that P-FAK levels were higher in $dy^W$ fetuses (Fig 5O), which is consistent with *Tmem182* down-regulation. Simultaneously, we observed an increase in *Itgb1* expression in E17.5 $dy^W$ muscle fibers compared with WT in our RNAseq data (Fig S3C). This aligns with previous reports (Packer & Martin, 2021), indicating that the absence of functional *Lama2* disrupts the stability of integrin $\beta$1, the $\beta$ chain of the $\alpha7\beta1$ integrin, a key laminin 211 transmembrane receptor.

### Cell cycle defects, increased DNA damage, and altered redox regulation are present at the onset of LAMA2-CMD

Considering that our data using the in vitro C2C12 model (Fig 1) and the RNAseq analysis of isolated E17.5 muscle fibers (Fig 4) suggest that pathways other than muscle differentiation are implicated in the onset of LAMA2-CMD, we examined cell proliferation, DNA damage, oxidative stress, and mitochondrial dysfunction in E17.5 whole muscles (Fig 6). mRNA expression analysis of the cell cycle inhibitors *Cdkn1a* (encoding p21) and *Cdkn1c* (encoding p57) showed a significant increase in fetal $dy^W$ muscle in comparison with WT, consistent with defective cell cycle regulation (Fig 6A). Similarly, levels of DNA damage were also higher in fetal muscles of

**Figure 4. RNA-sequencing analysis of muscle fibers from WT versus $dy^W$ fetuses at E17.5.**
**(A)** Muscle fibers were extracted from WT and $dy^W$ fetuses at E17.5, and RNA-sequencing analysis was performed. On the left is a schematic representation of the methodology used. On the right is the volcano plot of the differently expressed genes in $dy^W$ fetuses compared with WT. The down-regulated genes are represented in blue, the nonsignificant genes in gray, and the up-regulated ones in red. **(A, B, C)** Functional analysis of down-regulated (B) or up-regulated (C) genes in the RNAseq analysis described in (A). Counts indicate the number of genes found in each category. Class indicates the functional analysis class BP (biological process, circles), CC (cellular component, triangle), and MF (molecular function, square). Color indicates the scale for the $-\log_{10}$ (adjusted *P*-value). **(A, D)** Venn diagram comparing the differentially expressed genes (adjusted *P*-value of 0.01, $\log_2$ fold change of ±1.5) (described in (A)) with a previously published ChIP-seq database (10.17989/ENCSR000AHO, ENCODE), where C2C12 myoblasts and C2C12 myocytes (differentiated cells) were analyzed by histone ChIP-seq for different histone modifications. The presented Venn diagram specifically compares the histone H3 trimethylation on lysine 4 (H3K4me3) as a marker for active gene expression. **(D, E)** Functional analysis of the 191 genes with active expression in C2C12 myotubes but down-regulated in the RNAseq analysis of muscle fibers (identified in (D)). Counts indicate the number of genes found in each category. Class indicates the functional analysis class BP (biological process, circles), CC (cellular component, triangle), and MF (molecular function, square). Color indicates the scale for the $-\log_{10}$ (adjusted *P*-value). **(A, F, G, H)** Heatmaps of 20 representative genes (15 down-regulated and 5 up-regulated) from the Venn diagram analysis comparing the differentially expressed genes (adjusted *P*-value of 0.05, $\log_2$ fold change of ±1.5) of WT versus $dy^W$ muscle fibers from the RNAseq analysis in (A) with the following gene ontologies: (F) GO:0030036 Actin Cytoskeleton Organization and GO:0042692 Muscle Cell Differentiation; **(G)** GO:0007049 Cell Cycle and GO:0006281 DNA Repair; and **(H)** GO:0006979 Response to Oxidative Stress and GO:0007005 Mitochondrial Organization.

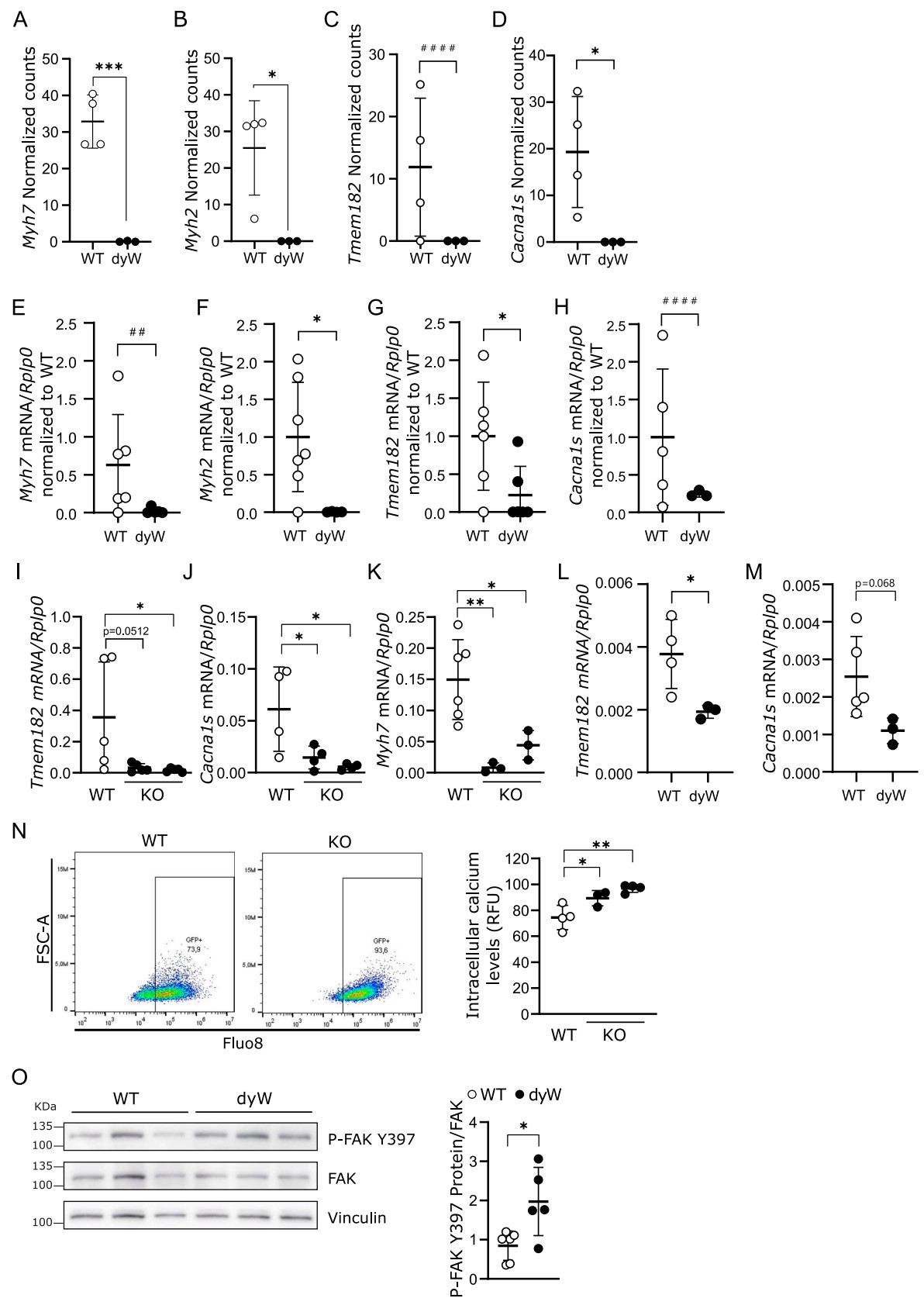

$dy^W$ compared with WT, as determined by the increase in the phosphorylation of histone H2AX (γH2AX) (Fig 6B). Despite the significant down-regulation observed for *Nqo1* gene expression in the RNAseq analysis (Fig 4H, Table S6), the same was not observed when analyzing NQO1 levels in fetal muscles from $dy^W$ and WT mice at E17.5 (Fig 6B). Likewise, no changes were observed in the levels of the OXPHOS complexes (Fig S4A). Considering that the onset of the disease was found to occur between E17.5 and E18.5 (Nunes et al, 2017), we asked whether the defect in antioxidant response and mitochondrial function could occur at E18.5. Our data showed that at E18.5, there were a significant decrease in the levels of NQO1 and an increase in complex I of the mitochondria (Fig 6C), whereas the remaining OXPHOS complexes were not altered (Fig S4B). These results are in accordance with our in vitro data (Fig 1), and further support the idea of an increase in oxidative stress at the onset of LAMA2-CMD.

## Discussion

Recent studies in mouse models of LAMA2-CMD by our laboratory and others have shown that molecular and cellular alterations associated with this disease are detected early in life (Nunes et al, 2017; Gawlik et al, 2019). Moreover, an increase in Janus kinase/ signal transducer and activator of transcription 3 (JAK/STAT3) signaling and a failure in normal muscle growth can be detected in fetal muscles of the $dy^W$ mouse model (Nunes et al, 2017), suggesting that disease onset occurs between E17.5 and E18.5. Using both an in vitro model of LAMA2-CMD and the well-established $dy^W$ mouse model for this disease, we have dissected the mechanisms underlying the onset of LAMA2-CMD. Our findings revealed a perturbation in cell proliferation (Figs 1 and 6A) and a profound defect in muscle differentiation, closely associated with cytoskeletal changes, as being at the core of the LAMA2-CMD onset in these two models (Figs 2–5 and S3). The timing of the defect in vivo (E17.5) correlates with a stage at which MuSCs have just entered their niche under the fully assembled basement membrane of myofibers (Kassar-Duchossoy et al, 2005; Nunes et al, 2017), which in the case of $dy^W$ fetuses (and LAMA2-CMD patients) does not contain laminin 211. Several studies have underscored the importance of laminin binding for cell proliferation and differentiation, in particular for stem cells (Li et al, 2002; Yap et al, 2019). Accordingly, MuSCs may

display a proliferation defect in the absence of *Lama2*, as indicated by our in vitro data (Fig 1C and D), and by our previous findings showing a reduction in the number of PAX7⁺ cells in $dy^W$ fetuses (Nunes et al, 2017). Importantly, our results also demonstrate that the absence of *Lama2* leads to drastic changes in muscle differentiation in vitro using both C2C12 cells and isolated primary fetal myoblasts (Figs 2 and S3A). These findings are consistent with observations in $dy^W$ embryonic stem cells (Kuang et al, 1998a) and correlate with a reduction in the myogenic regulatory factors *Myog* (Fig 3A) and MYF5 (Fig 3B and C), and an increase in the transcription factor NFIX (Fig 3C and D). These changes may indicate a failure to regulate the entry into fetal myogenesis (Messina et al, 2010; Ribeiro et al, 2023). Because NFIX expression should gradually increase during fetal development to assure a smooth transition from embryonic to fetal stages (Taglietti et al, 2016), it is plausible that this drastic alteration promotes cycles of regeneration and degeneration, exacerbating disease pathology. These cycles have been previously suggested in the context of α-sarcoglycan (*Sgca* null)– and dystrophin (*mdx*)-deficient dystrophic mice (Rossi et al, 2017). Even though muscle fiber generation is not compromised in the $dy^W$ fetuses (Nunes et al, 2017), our RNAseq analysis showed that these fibers are severely compromised, with a profound down-regulation of gene expression (Fig 4A). These results raise the possibility that treatment with histone deacetylase inhibitors, which promote gene expression, could have a positive effect on LAMA2-CMD, as previously shown in the context of other muscular dystrophies such as Duchenne muscular dystrophy (Minetti et al, 2006; Consalvi et al, 2011).

A comparison of the top down-regulated and up-regulated genes with a previously published H3K4me3 ChIP-seq database from C2C12 myoblasts and C2C12 myocytes allowed us to identify 191 genes expected to be active under differentiation conditions in C2C12 cells that were down-regulated in our RNAseq analysis of E17.5 $dy^W$ muscle fibers (Fig 4D, Table S1). These genes are involved in cell differentiation, including striated muscle cell differentiation, as well as cytoskeleton structure (Fig 4E). This further supports the hypothesis that a profound differentiation defect is at the core of LAMA2-CMD. Two key myosin heavy chain genes involved in muscle cell differentiation, *Myh7* and *Myh2*, were identified as being severely down-regulated (Fig 4A). MYH7 is a slow myosin that plays a critical role during embryonic and fetal muscle development. As development proceeds, other myosin chains start to be expressed,

**Figure 5.  Expression of genes linked to cell differentiation.**

**(A, B, C, D)** Normalized counts of genes selected from the analysis in Fig 4, which are related to cell differentiation (*Myh7* (A), *Myh2* (B), *Tmem182* (C), and *Cacna1s* (D)). n = 3–4 independent samples per genotype. **(E, F, G, H)** To validate the data shown in A–D, epaxial muscles were collected from WT and $dy^W$ fetuses at E17.5 and muscle fibers were separated. RNA was extracted from muscle fibers and cDNA synthesized. **(E, F, G, H)** Expression of genes selected from the analysis in Fig 4 was analyzed by qPCR (*Myh7* (E), *Myh2* (F), *Tmem182* (G), and *Cacna1s* (H)). Expression levels were normalized using the housekeeping gene *Rplp0*. n = 3–7 independent samples per genotype. **(I, J, K)** C2C12 WT cells and two independent *Lama2* knockout single-cell clones (KO) were cultured for 5 (I–J) or 14 d (K) to differentiate and then harvested. **(I, J, K)** RNA was extracted, and the expression of *Tmem182* (I), *Cacna1s* (J), and *Myh7* (K) was analyzed by qPCR. Expression levels were normalized using the housekeeping gene *Rplp0*. n = 3–6 independent samples per group were collected independently. **(L, M)** Primary fetal mouse myoblasts were isolated from WT and $dy^W$ fetuses at E16.5–E19.5, left to adhere and proliferate for 4 d, and then incubated with differentiation medium for an additional 3 d. **(L, M)** Cells were then harvested, and the expression of *Tmem182* (L) and *Cacna1s* (M) was analyzed. Expression levels were normalized using the housekeeping gene *Rplp0*. n = 3–5 fetuses per genotype. **(N)** WT and KO C2C12 were incubated with Fluo-8 and analyzed by flow cytometry. Representative flow cytometry plots (left panel) and respective quantification of intracellular calcium levels (right panel) are shown. RFU, relative fluorescence units. **(O)** Protein extracts from epaxial muscles of WT and $dy^W$ fetuses at E17.5 were analyzed by Western blot with anti-FAK Y397 and anti-FAK antibodies (left panel). Vinculin was used as the loading control. Densitometry analysis is shown on the right. n = 5–6 fetuses per genotype. **(A, B, C, D, E, F, G, H, I, J, K, L, M, N, O)** Statistical analysis was performed using an unpaired *t* test (*P*-value: \**P* < 0.05, \*\*\**P* < 0.001) or an F test to compare variances (*P*-value: ##*P* < 0.01, ####*P* < 0.0001) in (A, B, C, D, E, F, G, H, I, J, K, L, M, N, O) and with ordinary one-way ANOVA in (I, J, K, N). Data are represented as the mean ± SD. *P*-value: \**P* < 0.05, \*\**P* < 0.01. The ROUT method was used to identify outliers.

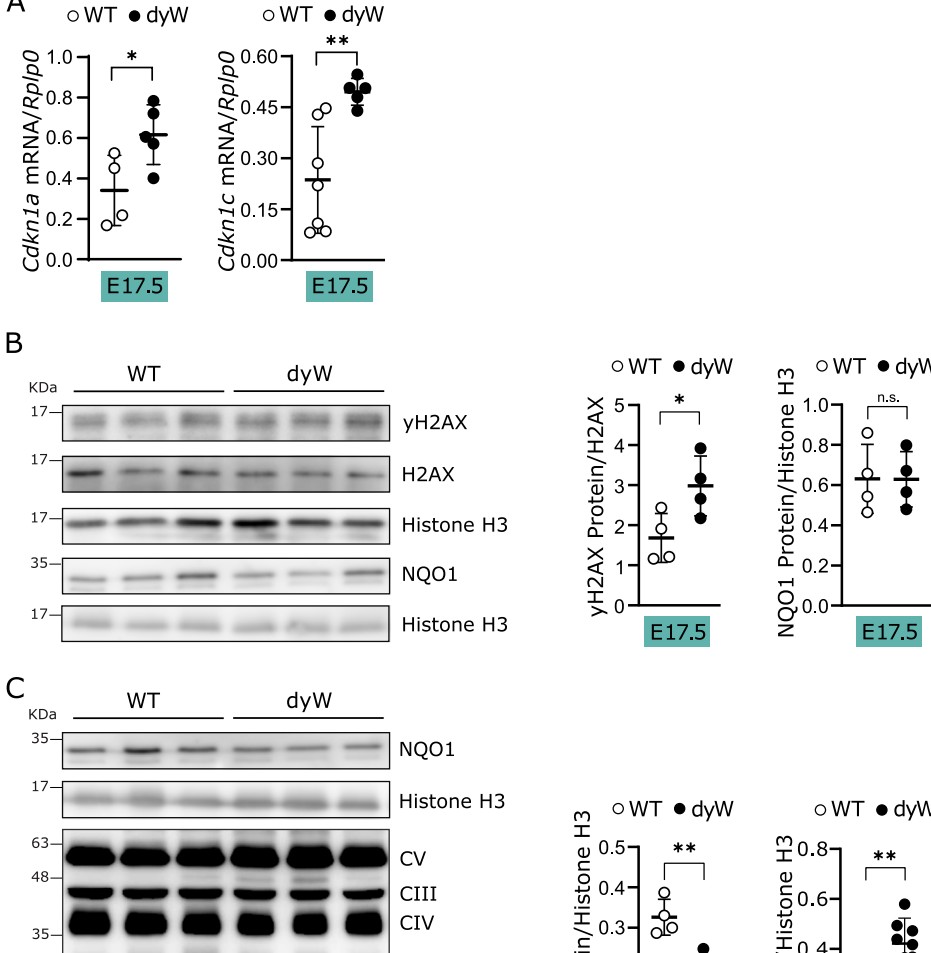

**Figure 6.** ***Lama2*** **deficiency leads to alteration in genes linked to cell cycle, and triggers DNA damage and oxidative stress.**
**(A)** Epaxial muscles were collected from WT and *dy^W* fetuses at E17.5, and RNA was extracted. The expression of *Cdkn1a* and *Cdkn1c* was analyzed by qPCR and normalized using the housekeeping gene *Rplp0*. n = 4–7 fetuses per genotype. **(B)** Epaxial muscles were collected from WT and *dy^W* fetuses at E17.5; the protein extracts were analyzed by Western blot with anti-γH2AX, anti-NQO1, and anti-H2AX antibodies (left panel). Histone H3 was used as the loading control. Densitometry analysis is shown on the right. n = 4 fetuses per genotype. **(C)** Epaxial muscles were collected from WT and *dy^W* fetuses at E18.5; the protein extracts were analyzed by Western blot with anti-NQO1 and anti-oxidative phosphorylation (CI, complex I; CII, complex II; CIII, complex III; CIV, complex IV; CV, complex V) antibodies (left panel). Histone H3 was used as the loading control. Densitometry analysis is shown on the right. n = 4–8 fetuses per genotype. **(A, B, C)** Statistical analysis was performed using an unpaired *t* test for (A, B, C). Data are represented as the mean ± SD. *P*-value: \**P* < 0.05, \*\**P* < 0.01. The ROUT method was used to identify outliers.

including the fast myosin MYH2, and are present from late fetal or early postnatal stages onward (Schiaffino et al, 2015). Thus, the reduced expression of *Myh7* and *Myh2* may compromise the differentiation and stability of *Lama2*-deficient myofibers (Figs 2 and 5). Moreover, this abnormal expression of myosin heavy chains may be related to the increased levels of NFIX (Fig 3C and D). This is consistent with previous studies showing that NFIX cooperates with SOX6 to repress the expression of MyHC-I (encoded by *Myh7*) during fetal muscle development (Taglietti et al, 2016).

Another two genes involved in myogenic differentiation, *Tmem182* and *Cacna1s* (Fig 4A), were identified as top hits in our RNAseq analysis, and these findings were validated both in vivo and in vitro (Fig 5). TMEM182 is a transmembrane protein with high expression in muscle and adipose tissues (Wu & Smas, 2008). Previous studies revealed that TMEM182 plays a role in myogenesis (Wu & Smas, 2008; Luo et al, 2021), where it negatively modulates ITGB1 by reducing its binding to laminin and by dampening ITGB1-

dependent FAK activity (Luo et al, 2021). The absence of *Lama2* leads to *Tmem182* down-regulation (Figs 4A and 5), and the consequent reduction in the TMEM182-ITGB1 interaction is perhaps linked to the observed increase in P-FAK levels (Fig 5M). Even though FAK activation typically promotes differentiation and fusion (Graham et al, 2015), the profound alterations observed in the absence of *Lama2* (Fig 4A) align with the presence of aberrant myotubes. One such alteration could be linked with the regulatory mechanisms maintaining intracellular calcium levels, as suggested by the significant decrease in *Cacna1s* and an increase in intracellular calcium levels (Figs 4A and 5). Importantly, increased cytosolic calcium has been shown to be a feature of Duchenne muscular dystrophy (Allen et al, 2010). Moreover, *Cacna1s* has been studied in the context of muscle differentiation in C2C12 cells (Huang et al, 2019; Qiu et al, 2022) and calcium deregulation has been associated with LAMA2-CMD disease pathology (de Oliveira et al, 2014). Down-regulation of *Cacna1s* has also been shown to occur in C2C12 cells treated with FGF9 (Huang

et al, 2019), and correlated with defective differentiation. In contrast to our findings, this study shows that FGF9 leads to a shift toward a more proliferative phenotype (Huang et al, 2019). In another study, the increase in the mRNA expression of *Cacna1s* and other calcium-related genes was found to coincide with that of myogenic factors such as *Myf5*, *Myod*, and *Myog* (Qiu et al, 2022). In keeping with these results, we showed that *Lama2* deficiency leads to a simultaneous decrease in the expression of *Myog* (Fig 3A), MYF5 (Fig 3B and C), and *Cacna1s* (Fig 5J). Finally, our in vivo findings from the RNAseq analysis (Fig 4A), and subsequent validation (Fig 5H), support the idea that *Cacna1s* expression is required not only for muscle differentiation, but also for myofiber maturation (Dos Santos et al, 2023).

In addition to perturbations in proliferation and a dramatic impairment in muscle differentiation, we have identified DNA damage and oxidative stress as hallmarks of the disease onset, both in in vitro (Fig 1) and in vivo (Figs 4 and 6). Moreover, our in vitro data (Fig S2C) and previous in vivo findings (Nunes et al, 2017) suggest that even though apoptosis is a hallmark of LAMA2-CMD and its inhibition improved the outcome of the disease (Girgenrath et al, 2004), it seems to be a consequence of the disease rather than a primary cause. This result is consistent with the absence of apoptotic fibers observed in the $dy^{3K}/dy^{3K}$ mouse model of LAMA2-CMD at E18.5 (Gawlik et al, 2019). DNA damage and oxidative stress could also influence muscle differentiation, as demonstrated by studies indicating that both DNA damage (Puri et al, 2002) and increased expression of HO-1 (Kozakowska et al, 2012) can inhibit C2C12 differentiation, the former by inhibiting muscle-specific genes (Puri et al, 2002) and the latter by targeting striated muscle–specific miRNAs, called myomiRs (Kozakowska et al, 2012). In addition, the decreased levels of reduced gluta-thione observed in our in vitro model (Fig 1E) align with the previously described aged MuSCs (Benjamin et al, 2023). This may provide an explanation for the reduction of the MuSC pool observed in $dy^W$ fetuses (Nunes et al, 2017). Our data also suggest that *Lama2* deficiency triggers DNA damage, in both the in vitro model (Fig 1M) and E17.5 fetuses in vivo (Figs 4G and 6B). This occurs before the observed changes in oxidative stress and mitochondrial composition, which occur at E18.5 (Fig 6C). Therefore, it is unlikely that the increase in DNA damage is a consequence of increased ROS levels, but rather an independent mechanism that can have an important contribution to the disease. Accordingly, DNA repair is decreased in differentiated myoblasts/myotubes, in particular base excision repair, a DNA repair mechanism that is the first line of response against DNA oxidation (Narciso et al, 2007). In the absence of *Lama2*, this might be exacerbated by the down-regulation of several DNA repair genes, including key genes involved in homologous recombination (e.g., *Brca2*, *Rad51c*, *Mdc1*, *Exo1*, *Mre11*) (Fig 4G, Table S5), a central pathway in DNA double-strand break repair. The down-regulation of DNA repair genes and the differentiation-related reduction in base excision repair are likely to contribute to the increase in DNA damage. This is further promoted by the later accumulation of ROS, contributing to myofiber degeneration. In addition, the link between defective DNA repair/cell cycle and oxidative stress–associated mitochondrial dysfunction may also be potentiated by the increased expression of *Cdkn1a* in the absence of *Lama2* (Fig 6A). *Cdkn1a* encodes the key cell cycle regulator p21, and increased levels of

p21 lead to mitochondrial dysfunction, DNA damage, and defective skeletal muscle (Englund et al, 2023). Possible mechanisms associated with the increase in oxidative stress may be the increase in mitochondrial complex I (Figs 1H and 6C) and changes in intracellular calcium (Fig 5L), because complex I has also been implicated in the regulation of calcium transport (Urra et al, 2017; Balderas et al, 2022). Moreover, mutations in complex I have also been associated with mitochondrial branching, as observed in the context of Leigh syndrome, a mitochondrial disorder (Meshrkey et al, 2021). This indicates that defective complex I regulation may have a pleiotropic effect, also leading to increased mitochondrial branching in *Lama2*-deficient myoblasts (Fig 1I–K). In vivo, an increase in complex I and a reduction in the levels of the anti-oxidant protein NQO1 are not evident at E17.5, but rather at E18.5 (Fig 6B and C), suggesting that they may be a consequence of an earlier dysfunction. Nonetheless, our findings support the notion that antioxidant treatment could be a potential therapeutic approach for LAMA2-CMD, as proposed by other studies in postnatal stages (Harandi et al, 2020).

The overall dramatic impact caused by the absence of *Lama2* leads to profound alterations in important signaling pathways that are involved in the communication between extra- and intracellular environments, such as FAK (Fig 5M) and JAK/STAT3 (Nunes et al, 2017). Another possibility is that faulty laminin 211–integrin binding to the $\alpha7\beta1$ integrin and/or dystroglycan alters cellular biomechanical properties, with potentially wide-ranging effects, including changes in gene expression (Goult et al, 2022). Several studies have shown that alterations in ECM structure and rigidity (i.e., ECM stiffness) can influence a variety of cellular processes (Handorf et al, 2015). For example, we detected a collagen-enriched ECM signature in $dy^W$ fetuses (Fig 4C), which can impair proliferation (Lacraz et al, 2015), impact nuclear stability and trigger DNA damage (Cho et al, 2019; Deng et al, 2020), and result in changes in mitochondrial function and fusion (Chen et al, 2021; Tharp et al, 2021; Cai et al, 2023). These modifications in ECM composition likely represent the initial signs of fibrosis, which characterizes later stages of LAMA2-CMD (Yurchenco et al, 2018; Gawlik & Durbeej, 2020), but not its onset (Nunes et al, 2017).

In this study, we present the first evidence of what molecular and cellular processes underlie the onset of LAMA2-CMD in mouse models of the disease. Although previous studies have reported proliferation and differentiation defects, along with increased oxidative stress and metabolic alterations (de Oliveira et al, 2014; Fontes-Oliveira et al, 2017; Nunes et al, 2017; Yurchenco et al, 2018; Gawlik et al, 2019; Kölbel et al, 2019; Gawlik & Durbeej, 2020; Harandi et al, 2020; Martins et al, 2021), as important hallmarks for LAMA2-CMD disease pathology, we have, for the first time, linked some of these changes to the disease onset during fetal development. Specifically, we have shown that *Lama2*-deficient myoblasts cultured in vitro increase NFIX levels and fail to up-regulate Myogenin and Myomaker, leading to an inability to fuse. Moreover, we showed that the expression of key genes associated with muscle differentiation, including *Myh2*, *Myh7*, *Tmem182*, and *Cacna1s*, is severely impaired in the absence of *Lama2*, as soon as laminin-$\alpha2$ assumes its role as the central laminin isoform in the basement membrane of muscle fibers.

Our results provide an important framework for studies aiming to therapeutically target the primary defect of this disease, which can hopefully increase the lifespan and quality of life of LAMA2-CMD

patients. In addition, because a significant number of alterations appear to be shared among various muscular dystrophies, these findings could offer valuable insights into the study of other less understood muscular dystrophies.

# Materials and Methods

### Animals

Mice were bred and maintained under specific pathogen-free (SPF) conditions at the Instituto Gulbenkian de Ciência (IGC), and heterozygous crossings were performed at FCUL. All experimental protocols were approved by the Ethics Committee of IGC, the organization responsible for animal welfare (*Órgão Responsável pelo Bem-estar dos Animais* [ORBEA]) at IGC and at FCUL, and subsequently licensed by the Portuguese National Entity (*Direcção Geral de Alimentação e Veterinária*) (license 0421/000/000/2020 and 0421/000/000/2022). Experimental procedures were performed according to the Portuguese (Portaria no 1005/92, Decreto-Lei no 113/2013, and Decreto-Lei no 1/2019) and European (Directive 2010/63/EU) legislations concerning housing, husbandry, and animal welfare. Heterozygous $dy^W$ C57BL/6 mice (gift from Eva Engvall via Dean Burkin; the University of Nevada, Reno, NV, USA) were crossed to obtain homozygous $dy^W$ mutants and WT fetuses (Kuang et al, 1998b). Pregnant females were anesthetized with isoflurane and euthanized by cervical dislocation, and the uterine horns were removed and placed in ice-chilled PBS. Fetuses were removed from the uterine horns, maintained in ice-chilled PBS, and decapitated. Tails from fetuses were collected for genotyping. For tail lysis, 25 mM NaOH/0.2 mM EDTA was added to the samples and they were placed in a thermocycler at 95°C for 30 min; then, 40 mM Tris–HCl, pH 5.5, was added to the mixture in a 1:1 ratio and centrifuged for 3 min. For PCR, 1 $\mu$l of undiluted mixture was used per reaction. All PCRs were performed using Xpert Fast Hotstart DNA Polymerase according to the manufacturer's instructions. The primers used are listed in the Key Resources Table (*Oligonucleotides—DNA genotyping*). Epaxial muscles were isolated from E17.5 and E18.5 WT and $dy^W$ fetuses and used in the different analyses or stored at −80°C for long-term preservation.

### Cell lines

C2C12 myoblast cell lines were grown in Dulbecco's modified Eagle's medium (DMEM), supplemented with 10% FBS and 1% of an antibiotic mixture: penicillin (10,000 U/ml) and streptomycin (10 mg/ml) (complete or proliferation medium). The cell lines were maintained in culture at 37°C, 5% $CO_2$, and constant humidity. When 70% of confluency was reached, trypsin–EDTA or TrypLE Express Enzyme was used to passage the cells.

### Generation of *Lama2* KO C2C12 cell lines

The in vitro model for LAMA2-CMD was established through the deletion of the *Lama2* gene from C2C12 cells by CRISPR/Cas9 (knockout C2C12 cell lines) using two gRNAs targeting exons 4

and 9 of *Lama2* cloned in pRP[CRISPR]-Puro-hCas9-U6 (Key Resources Table *Oligonucleotides—gRNA oligos and cloning oligos*). Transfection of C2C12 cells was performed using the Lipofectamine 3000 transfection reagent, according to the manufacturer's instructions. Approximately 48 h after transfection, selection was performed using 3 μg/ml puromycin for 48 h. Upon selection, single-cell clones were isolated by limiting dilution (8 cells/ml). Alternatively, C2C12 cells were cotransfected with the plasmids carrying the gRNA and a GFP-expressing plasmid. In this second approach, GFP-positive cells were individually isolated through FACS (FACSAria III [BD Biosciences]). In both approaches, isolated clones were expanded in complete medium with 20% FBS. qPCR was performed to evaluate the *Lama2* deletion in C2C12 single-cell clones (Key Resources Table *Oligonucleotides—qPCR oligos*), and Sanger sequencing (STAB VIDA) was performed to confirm the gene editing. To confirm the presence of deletions, a PCR was also performed (see Key Resources Table *Oligonucleotides—gRNA oligos and cloning oligos* for primers and the animal section on methods for PCR protocol). Transfection and clonal selection were recurrently performed over time, taking into account the characteristics associated with *Lama2* deletion.

### Differentiation assay

WT and *Lama2* knockout C2C12 cells were plated (day −1) in complete DMEM at a density of 50,000 cells per well in a 24-well plate with coverslips and left to differentiate until day 5 or 14. Images were acquired on days 0, 2, 5, 7, 12, and 14 using Optika IM-3LD4 with LED fluorescence under a 20x objective lens coupled with a Canon EOS M200 camera. On day 5 or day 14, cells were fixed for immunofluorescence assay or collected for RNA extraction.

### Proliferation assay

To analyze the proliferation of the different C2C12 cell lines, a resazurin assay was performed. For that, on the day before the experiment, WT and *Lama2* knockout C2C12 cells were plated in a 96-well plate (5,000 cells per well). On the next day (day 0), cells were incubated in 1x resazurin solution in complete DMEM for 1 h 20 min. Fluorescence (excitation filter 531/40 nm; emission filter 595 nm) was measured using a Victor 3V plate reader (PerkinElmer) on days 0, 1, 2, and 5. The fluorescence levels were normalized to control and to day 0 for each cell line.

### Cell cycle analysis

To evaluate the cell cycle progression of the different C2C12 cell lines, the cells and growth media were collected and washed with 1x PBS. They were subsequently resuspended in PBS, followed by the addition of ice-cold absolute ethanol drop by drop. Cells were fixed at −20°C for at least 24 h. Immediately before the analysis by flow cytometry, cells were centrifuged, the ethanol was discarded, and the cells were resuspended in 1x PBS with 5 mg/ml RNase for 5 min at room temperature (RT). After incubation, the cells were centrifuged and resuspended in 1x PBS with 50 $\mu$g/ml of propidium iodide (PI). Flow cytometry analysis was performed using BD FACSCalibur Flow Cytometer (BD Biosciences).

## Glutathione assay

WT and *Lama2* knockout C2C12 cells were plated in a 24-well plate with a confluency of 70–90%. To measure the levels of reduced glutathione, cells were washed with serum-free DMEM without phenol red and then with FluoroBrite DMEM. Cells were washed and then incubated with 40 $\mu$M of monochlorobimane diluted in FluoroBrite DMEM for 30 min. Fluorescence was measured using a plate reader (Spark 10M; TECAN) with an excitation wavelength of 390 nm and an emission wavelength of 490 nm. For each well, 36 measurements of different sections of the well were obtained and then normalized for the protein levels of each well. For protein quantification, cells were recovered and lysed in lysis buffer (50 mM Tris, pH 6.8, 2% SDS, and 10% glycerol), and then, protein levels were measured using the BCA quantification method.

## Fractionation assay

A cell fractionation protocol was used to separate nuclear and cytoplasmic fractions (Méndez & Stillman, 2000). WT and *Lama2* knockout C2C12 cells were collected and washed in PBS, then resuspended in ice-cold fractionation buffer (10 mM Hepes, pH 7.9, 10 mM KCl, 1.5 mM MgCl$_2$, 0.34 M sucrose, 10% glycerol, 0.075% Triton X-100, 1 mM DTT, and a protease inhibitor cocktail), and incubated on ice for 5 min. The samples were centrifuged for 5 min at 1,300*g* at 4°C. The supernatant containing the cytoplasmic proteins was cleared by centrifugation for 20 min at 20,000*g* at 4°C and then resuspended in 2x SDS–PAGE sample buffer (20% glycerol, 4% SDS 100 mM, Tris, pH 6.8, 0.2% bromophenol blue, and 100 mM DTT) in a 1:1 ratio. The pellet containing the nuclear proteins was washed twice with ice-cold fractionation buffer, centrifuged for 5 min at 1,300*g* at 4°C, and resuspended in 2x SDS–PAGE sample buffer. Protein concentration was measured with NanoDrop 1000 at 280 nm.

## Isolation, culture, and differentiation of mouse primary fetal myoblasts

Mouse primary fetal myoblasts were isolated as previously described (Pimentel et al, 2017), with minor modifications. Briefly, skeletal muscles from WT and *dy*$^W$ fetuses at E16.5–E19.5 were extracted, minced, and then digested with a digestion mix (5 mg/ml of collagenase type V and 3.5 mg/ml of dispase in DPBS) for 1 h at 37°C with vigorous agitation. After the digestion was complete, dissection medium (10% FBS and 1% of an antibiotic mixture: penicillin [10,000 U/ml] and streptomycin [10 mg/ml] in GlutaMAX-supplemented IMDM) was added, and cells were centrifuged to pellet the remaining tissue and fat. The supernatant was recovered and centrifuged to pellet the cells, which were then resuspended in fresh dissection media. Cells were filtered through a 40-$\mu$m cell strainer directly into a 100-mm dish and incubated at 37°C with 5% CO$_2$ for 1 h to allow the fibroblasts to adhere. After this preplating step, the supernatant was collected and centrifuged to pellet the cells. The cells obtained were resuspended in growth medium (20% FBS, 1% of an antibiotic mixture: penicillin [10,000 U/ml] and streptomycin [10 mg/ml], and 1% chicken embryo extract in GlutaMAX-supplemented IMDM), counted, and plated in 24-well plates coated with 50 $\mu$g/ml of collagen I at a density of 50,000 cells per well. After 4 d in growth

medium, differentiation medium (2% horse serum and 1% of an antibiotic mixture: penicillin [10,000 U/ml] and streptomycin [10 mg/ml] in GlutaMAX-supplemented IMDM) was added and cells were cultured for 3 d. Cells were then either processed for qPCR analysis or fixed for immunofluorescence analysis.

## RNA extraction and qPCR

Epaxial muscles collected from E17.5 and E18.5 fetuses, C2C12 cells, or primary fetal myoblasts were lysed using the tripleXtractor reagent and a tissue homogenizer (Retsch MM400 TissueLyser). The RNA was extracted according to the manufacturer's instructions. RNA concentrations and quality were determined with NanoDrop 1000. Xpert cDNA Synthesis Kit was used to synthesize complementary DNA (cDNA) according to the manufacturer's instructions. The cDNA obtained was stored at –20°C until further analysis by qPCR. qPCRs were performed with iTaq Universal SYBR Green Supermix or Xpert Fast SYBR (Uni) Blue, according to the manufacturer's instructions, using CFX96 Real-Time PCR Detection System (Bio-Rad). Transcript levels were normalized against *Rplp0* expression, and the fold change was calculated using the ΔΔCt method. Primers used are listed in Key Resources Table *Oligonucleotides—qPCR oligos*.

## Analysis of mitochondrial morphology

WT and *Lama2* knockout C2C12 cell lines were incubated for 30 min at 37°C with 200 nM of MitoTracker Deep Red FM. Cells were then washed twice with PBS 1x and fixed with 4% PFA for 10 min at RT. After fixation, cells were permeabilized with 0.1% Triton X-100 for 5 min, washed with PBS 1x, and incubated with blocking solution (1% BSA, 1% goat serum, and 0.05% Triton X-100 in PBS) for 1 h at RT. Cells were then incubated with Alexa Fluor 488 Phalloidin for 30 min and washed three times with PBS. Nuclei were stained with DAPI for 30 s. Samples were mounted using Mowiol/DABCO mounting medium and viewed in a Leica DMI4000B TCS SP5 fluorescence microscope equipped with a 63x objective lens and coupled to a Leica DFC365 FX 1.4 MP CCD camera. Mitochondrial branching was analyzed using the plugin Mitochondria Analyzer on ImageJ/Fiji following a previously described pipeline (Chaudhry et al, 2020). Phalloidin staining was used to assure that all the MitoTracker signal detected was located in the cytoplasm of the cells.

## Quantification of the number of mitochondria per cell

WT and *Lama2* knockout C2C12 cell lines were harvested, and total DNA was extracted. Quantification of mitochondrial numbers was done using the protocol previously described by Quiros et al (2017).

## Western blot

To extract the protein from muscle and C2C12 cells, samples were collected and resuspended in 2x SDS–PAGE sample buffer. Then, cells and tissues were homogenized with the tissue homogenizer (MM400 TissueLyser; Retsch), further incubated with Benzonase for 20 min, when required, heated at 50°C for 10 min, and centrifuged at a maximum speed for 5 min. Protein quantification was measured with

NanoDrop 1000 at 280 nm. Protein extracts were separated with 10, 12, or 15% SDS–polyacrylamide gel electrophoresis in running buffer (3.02$g$ Tris base, 14.42$g$ glycine, and 1$g$ SDS in 1 liter distilled water) using the Mini-PROTEAN Tetra electrophoresis system (Bio-Rad). Then, proteins were transferred to PVDF membranes on Mini Trans-Blot Cell (Bio-Rad) with chilled transfer buffer (5.82$g$ Tris and 2.93$g$ glycine in 1 liter of distilled water) and blocked with 5% milk in TBST (20 mM Tris, 150 mM NaCl, 0.1% Tween-20, and distilled water, pH 7.4–7.6), with agitation. Protein loading was verified by GelCode Blue Safe Protein Stain. Primary antibodies diluted in 2% BSA in TBST and 0.02% sodium azide were incubated overnight at 4°C with agitation (antibodies and dilutions used are listed in Key Resources Table). Secondary antibodies coupled with horseradish peroxidase (HRP) diluted in 5% powdered milk in TBST were incubated for 1 h at RT, after being previously washed in TBST. Signal detection was obtained with SuperSignal West Pico Chemiluminescent Substrate HRP, and images were acquired using Amersham Imager 680 RGB (GE Healthcare). Quantifications were performed in Fiji software.

## Immunofluorescence

Primary fetal myoblasts were cultured as described earlier (*Isolation, culture, and differentiation of mouse primary fetal myoblasts*). WT and *Lama2* knockout C2C12 cell lines were plated in a 24-well plate with coverslips (50,000 cells per well). Cells were washed with 1x PBS and fixed for 10 min with 4% PFA in PBS, at RT. After washing with PBS, cells were permeabilized in 0.1% Triton X-100 in PBS for 5 min and washed with PBS. Then, cells were blocked with blocking solution (1% BSA, 1% goat serum, and 0.05% Triton X-100 in PBS) for 1 h at RT and incubated with the primary antibody (see Key Resources Table—*Antibodies for Immunofluorescence*) diluted in blocking solution overnight at 4°C. The next day, cells were washed with blocking solution and incubated with the secondary antibody (see Key Resources Table—*Antibodies for Immunofluorescence*) diluted in blocking solution for 1 h at RT. Nuclei were stained with DAPI for 30 s. Samples were mounted using Mowiol/DABCO mounting medium. Images were acquired using an Olympus BX60 fluorescence microscope equipped with a 20x objective lens and coupled to a Hamamatsu Orca R2 cooled monochromatic CCD camera.

## Measurement of intracellular calcium

WT and *Lama2* knockout C2C12 cell lines were plated in a 24-well plate (40,000 cells/well). On the next day, cells were incubated with 4 $\mu$M of Fluo-8 probe at 37°C for 1 h or left untreated. After the incubation period, cells were washed with 1x PBS and then resuspended in 1x PBS. Cells were analyzed using CytoFLEX Flow Cytometer (Beckman Coulter).

## RNA sequencing and analysis of muscle fibers

Epaxial muscles from WT and $dy^W$ fetuses at E17.5 were collected and digested in 37°C digestion buffer (6 U/ml dispase II and 1 U/ml collagenase A in DMEM complete medium) with strong agitation for 45 min at 37°C. After incubation, the digestion buffer was inactivated with ice-cold DMEM. To recover the fibers, the mixture was passed through a 70-$\mu$m strainer. Fibers were collected from the strainer, washed with 1x PBS, and resuspended in RTL lysis buffer (for sequencing) or tripleXtractor reagent (for RNA extraction and qPCR). Samples in lysis buffer were sent to the Genomics Facility of Instituto Gulbenkian de Ciência (IGC) to proceed with library preparation and next-generation sequencing. Briefly, the SMART-seq2 protocol was used to generate full-length cDNAs and sequencing libraries directly from lysed muscle fibers. The Nextera library preparation protocol (Nextera XT DNA Library Preparation Kit, Illumina) was used for library preparation, including cDNA "tagmentation," PCR-mediated adaptor addition, and library amplification. The libraries were sequenced using NextSeq 2000 P2 Reagents (100 cycles) (Illumina). Sequencing data were extracted in FastQ format, resulting on average in ~30 × 10$^6$ reads per sample, and analyzed using the Galaxy Europe platform (Afgan et al, 2018; Galaxy Community et al, 2022), following a previously reported workflow (Batut et al, 2021). The quality of FastQ reads was analyzed with *FastQC*, and the reads were then trimmed and filtered using *Cutadapt* and finally aligned against the mouse reference genome GRCm39 using *RNA STAR*. Read summarization was performed by assigning uniquely mapped reads to genomic features using *FeatureCounts*, and then, differential gene expression analysis comparing WT and $dy^W$ muscle fibers was performed using DESeq2. Functional analysis was performed with the gProfiler platform (https://biit.cs.ut.ee/gprofiler/page/citing), and bubble plot representation was performed using SRplot (Tang et al, 2023).

ChIP-seq data for C2C12 myoblast and myocyte (H3K4me3 peaks) were obtained from the ENCODE project (https://www.encodeproject.org/help/citing-encode/). Peak/gene association was performed using ChIPseeker (Yu et al, 2015).

Gene lists of GO:0030036, GO:0042692, GO:0006281, GO:0006979, GO:0007005, and GO:0007049 were obtained from Mouse Genome Informatics (MGI) Web site (https://www.informatics.jax.org/mgihome/other/citation.shtml). The Venn diagram analysis was performed using Platform Bioinformatics & Evolutionary Genomics (https://bioinformatics.psb.ugent.be/webtools/Venn/), whereas the heatmaps were generated using the Morpheus platform (https://software.broadinstitute.org/morpheus/).

## Quantification and statistical analysis

ImageJ/Fiji software was used to analyze immunofluorescence images and perform densitometry analysis of Western blots. Statistical analysis was conducted using GraphPad Prism 9 software. All distributed data are displayed as means ± SD of the mean unless otherwise noted. Measurements between two groups were performed with an unpaired $t$ test. Groups of three or more were analyzed by ordinary one-way ANOVA with Dunnett's multiple comparisons test and two-way ANOVA with Tukey's multiple comparisons test, depending on the number of variables analyzed. Statistical parameters for each experiment can be found within the corresponding figure legend. Serial cloner 2.6.1, SnapGene software, and Synthego Performance Analysis, ICE Analysis (2019.v3.0), were used for sequencing analysis.

## Reagents, antibodies, and resources

All reagents, antibodies, and resources are described in the Key Resources Table.

**Key Resources Table.**

| Reagent or resource | Source | Identifier |
|---|---|---|
| Antibodies for Western blot | | |
| GAPDH (1:30,000) Mouse | Proteintech | Cat# 60004-1-Ig, (RRID:AB_2107436) |
| GAPDH (1:2,000) Rabbit | Cell Signaling | Cat# 2118, (RRID:AB_561053) |
| Goat Anti-Mouse IgG-HRP (1:5,000) | Jackson ImmunoResearch Europe | Cat# 115-035-003 (RRID:AB_10015289) |
| Goat Anti-Rabbit IgG-HRP (1:5,000) | Jackson ImmunoResearch Europe | Cat# 111-035-003 (RRID:AB_2313567) |
| gH2AX S139 (1:1,000) Mouse | Millipore | Cat# 05-636 (RRID:AB_309864) |
| H2AX (1:1,000) Rabbit | Bethyl Laboratories | Cat# A300-082A (RRID:AB_203287) |
| Histone H3 (1:2,000) Rabbit | Cell Signaling | Cat# 9715, (RRID:AB_331563) |
| HO-1 (1:1,000) Rabbit | Enzo Life Sciences | Cat# SPA-896 (RRID:AB_2118666) |
| MAP LC3 α/β G-4 (1:1,000) Mouse | Santa Cruz Biotechnology | Cat# sc-398822 (RRID:AB_2877091) |
| MYF5 (1:1,000) Mouse | Santa Cruz Biotechnology | Cat# sc-302 (RRID:AB_631994) |
| NFIX (1:1,000) Rabbit | Novus Biologicals | Cat# NBP2-15039 (RRID:AB_2891313) |
| NQO1 D6H3A (1:1,000) Rabbit | Cell Signaling | Cat# 62262 (RRID:AB_2799623) |
| OXPHOS (1:2,000) Mouse | Abcam | Cat# ab110413 (RRID:AB_2629281) |
| SLC7A11/xCT (1:1,000) Rabbit | Proteintech | Cat# 26864-1-AP (RRID:AB_2880661) |
| Vinculin (1:1,000) Mouse | Abcam | Cat# ab18058, (RRID:AB_444215) |
| Antibodies for immunofluorescence | | |
| 53BP1 (1:100) Rabbit | Cell Signaling | Cat# 4937 (RRID:AB_10694558) |
| Alexa Fluor 568 Anti-Mouse IgG (1:500) | Molecular Probes | Cat# A11019 (RRID:AB_143162) |
| Alexa Fluor 488 Anti-Rabbit IgG (1:500) | Molecular Probes | Cat# A11070 (RRID:AB_142134) |
| Alexa Fluor 488 Phalloidin (1:200) | Thermo Fisher Scientific | Cat# A12379 |
| MYF5 (1:150) Mouse | Santa Cruz Biotechnology | Cat# sc-302 (RRID:AB_631994) |
| Myosin heavy chain (1:100) Mouse | DSHB | Cat# MF 20 (RRID:AB_2147781) |
| NFIX (1:200) Rabbit | Novus Biologicals | Cat# NBP2-15039 (RRID:AB_2891313) |
| Reagents | | |
| Benzonase | Millipore | Cat# 70746-3 |
| BSA, Fraction V | NZYtech | Cat# MB04601 |
| Bromophenol blue | AppliChem | Cat# 131165-0005 |
| Calcium chloride (KCl) | AppliChem | Cat# A3582 |
| Chloroform | Merck | Cat# 102445 |
| Collagen I | Sigma-Aldrich | Cat# C3867-1VL |
| Collagenase type A | Roche | Cat# 10103578001 |
| Collagenase type V | Sigma-Aldrich | Cat# C9263 |
| Dispase II | Sigma-Aldrich | Cat# D4693 |
| DMEM, high glucose | Biowest | Cat# L0103 |
| DMEM without phenol red | Gibco | Cat# 11594416 |
| DTT | Bio-Rad | Cat# 1610610 |
| EDTA | Merck | Cat# 108418 |
| Ethanol | Fisher Chemical | Cat# E/0600DF/17 |
| FBS | Biowest | Cat# S1560 |
| Fluo-8, AM | AAT Bioquest | Cat# 21080 |
| FluoroBrite DMEM | Gibco | Cat# A1896701 |
| GelCode Blue Safe Protein Stain | Thermo Fisher Scientific Pierce | Cat# PIER24594 |

| Reagent or resource | Source | Identifier |
|---|---|---|
| Glycerol | Sigma-Aldrich | Cat# G2025 |
| Glycine | Alfa Aesar | Cat# A13816.0E |
| Hepes | AppliChem | Cat# A1069 |
| Horse serum | GE Healthcare | Cat# 11581831 |
| IMDM, GlutaMAX-supplemented | Gibco | Cat# 31980022 |
| Isopropanol | Fisher Chemical | Cat# P/7500/17 |
| iTaq Universal SYBR Green Supermix | Bio-Rad | Cat# 1725120 |
| Lipofectamine 3000 | Thermo Fisher Scientific–Life Technologies | Cat# CMAX00003 |
| MitoTracker Deep Red FM | Invitrogen | Cat# M22426 |
| Monochlorobimane | Sigma-Aldrich | Cat# 69899 |
| Penicillin/streptomycin | GRiSP | Cat# GTC05.0100 |
| Dulbecco's phosphate buffered saline (DPBS) 10X w/o calcium w/o magnesium | Biowest | Cat# X0515 |
| Propidium iodide | AppliChem GmbH | Cat# A2261.0025 |
| Puromycin | Gibco | Cat# A1113803 |
| Resazurin sodium salt | Alfa Aesar | Cat# B21187.03 |
| Sodium chloride | Merck | Cat# 106404 |
| SDS | Merck | Cat# 113760 |
| Sodium hydroxide (NaOH) | Merck | Cat# 106462 |
| Sucrose | Merck | Cat# 107687 |
| SuperSignal West Pico Chemiluminescent Substrate HRP | Thermo Fisher Scientific Pierce | Cat# PIER34580 |
| tripleXtractor reagent | GRiSP | Cat# GB23.0200 |
| Tris base | Fisher BioReagents | Cat# BP152-5 |
| TrypLE Express Enzyme | Gibco | Cat# 12605010 |
| Trypsin–EDTA | GRiSP | Cat# GTC02.0100 |
| Tween-20 | Merck | Cat# 822184 |
| Xpert cDNA Synthesis Kit | GRiSP | Cat# GK80.0100 |
| Xpert Fast Hotstart 2X Mastermix with Dye | GRiSP | Cat# GE45.5001 |
| Xpert Fast SYBR (Uni) Blue | GRiSP | Cat# GE22.5100 |
| Deposited data | | |
| RNA sequencing | GEO database | GSE253680 |
| Experimental models: cell lines | | |
| Mouse: C2C12 | Leibniz Institute DSMZ | Cat# ACC 565 (RRID:CVCL_0188) |
| Mouse: *Lama2* KO C2C12 | This study | |
| Experimental models: organisms/strains | | |
| Mouse: *B6.129S1(Cg)-Lama2$^{tm1Eeng}$/J* | gift from Eva Engvall via Dean Burkin; the University of Nevada, Reno, NV, USA | |
| Oligonucleotides—DNA genotyping | | |
| Oligo | Sequence | Reference |
| *dy$^W$* 1 | 5'-ACTGCCCTTTCTCACCCACCCTT-3' | Nunes et al (2017) |
| *dy$^W$* 2 | 5'-GTTGATGCGCTTGGGACTG-3' | Nunes et al (2017) |
| *dy$^W$* 3 | 5'-GTCGACGACGACAGTACTGGCCTCAG-3' | Nunes et al (2017) |
| Oligonucleotides—gRNA oligos and cloning oligos | | |

**Continued**

| Reagent or resource | | | Source | Identifier |
|---|---|---|---|---|
| Gene | Name | Exon | Sequence | Brand |
| *Lama2* | gRNA43 | 4 | 5'-GGCTGCCTTCACAATTACGT-3' | VectorBuilder |
| *Lama2* | gRNA167 | 9 | 5'-GATGAGAAATATGCCCAGCG-3' | VectorBuilder |
| *Lama2* | Forward primer | 4 | 5'-GTTTGCACTAATGCTGGACCC-3' | This study |
| *Lama2* | Reverse primer | 4 | 5'-GCTTTCATTAGGTCTCTGGCCT-3' | This study |
| *Lama2* | Forward primer | 9 | 5'-GTGTCTGGCAACATCGCAT-3' | This study |
| *Lama2* | Reverse primer | 9 | 5'-TGAAGCCCACTGTGACATTTCT-3' | This study |
| Oligonucleotides—qPCR oligos | | | | |
| Oligonucleotides | | | Sequence | References |
| *Rplp0 Fwd* | | | 5'-GCTTTCTGGAGGGTGTCCG-3' | This study |
| *Rplp0 Rev* | | | 5'-ACGCGCTTGTACCCATTGAT-3' | This study |
| *Lama2 Fwd* | | | 5'-TGAAAGCAAGGCCAGAAGTCA-3' | This study |
| *Lama2 Rev* | | | 5'-ACAAAACCAGGCTTGGGGAA-3' | This study |
| *Cdkn1c Fwd* | | | 5'-GCCAATGCGAACGACTTCTT-3' | This study |
| *Cdkn1c Rev* | | | 5'-ATCTCAGACGTTTGCGCGG-3' | This study |
| *Cdkn1a Fwd* | | | 5'-TGTCTGAGCGGCCTGAAGATT-3' | This study |
| *Cdkn1a Rev* | | | 5'-AAGACCAATCTGCGCTTGGAGT-3' | This study |
| *Bax Fwd* | | | 5'-AAACTGGTGCTCAAGGCCC-3' | This study |
| *Bax Rev* | | | 5'-TTGGATCCAGACAAGCAGCC-3' | This study |
| *Bcl2 Fwd* | | | 5'-AACAGGGAGATGTCACCCCTGG-3' | This study |
| *Bcl2 Rev* | | | 5'-AGCCTCCGTTATCCTGGATCC-3' | This study |
| *Myog Fwd* | | | 5'-GTCCCAACCCAGGAGATCAT-3' | This study |
| *Myog Rev* | | | 5'-CCACGATGGACGTAAGGGAG-3' | This study |
| *Tubb6 Fwd* | | | 5'-AAGAAGTACGTACCCAGGGC-3' | This study |
| *Tubb6 Rev* | | | 5'-CACCCGTCTGTCCGAAGAT-3' | This study |
| *Myl1 Fwd* | | | 5'-CGGAGTTTTCAAGCACGCAA-3' | This study |
| *Myl1 Rev* | | | 5'-TCTGCATGGTGGTAAGCTGG-3' | This study |
| *Myh7 Fwd* | | | 5'-CTCAAGCTGCTCAGCAATCTATTT-3' | Zhou et al (2010) |
| *Myh7 Rev* | | | 5'-GGAGCGCAAGTTTGTCATAAGT-3' | Zhou et al (2010) |
| *Myh2 Fwd* | | | 5'-GAATGCCTACGAGGAGTCTCT-3' | Emerald et al (2022) |
| *Myh2 Rev* | | | 5'-TTCTGCAATCTGTTCCGTGA-3' | Emerald et al (2022) |
| *Tmem182 Fwd* | | | 5'-TCTGGAAGTTCTGGTACACCAAT-3' | This study |
| *Tmem182 Rev* | | | 5'-CGTAGGAGGTGGAGTTGTGC-3' | This study |
| *Cacna1s Fwd* | | | 5'-TGAGCGAGATCGATGACCCA-3' | This study |
| *Cacna1s Rev* | | | 5'-ACGATCAGCAAAGCCACGTA-3' | This study |
| *Atg5_Fwd* | | | 5'-CTTGCATCAAGTTCAGCTCTTCC-3' | OriGene |
| *Atg5_Rev* | | | 5'-AAGTGAGCCTCAACCGCATCCT-3' | OriGene |
| *Atg7_Fwd* | | | 5'-CAGAAGAAGTTGAACGAGTA-3' | This study |
| *Atg7_Rev* | | | 5'-CAGAGTCACCATTGTAGTAAT-3' | This study |
| *Mymk_Fwd* | | | 5'-ATCGCTACCAAGAGGCGTT-3' | This study |
| *Mymk_Rev* | | | 5'-CACAGCACAGACAAACCAGG-3' | This study |
| *Mymx_Fwd* | | | 5'-GTTAGAACTGGTGAGCAGGAG-3' | Bi et al (2018) |
| *Mymx_Rev* | | | 5'-CCATCGGGAGCAATGGAA-3' | Bi et al (2018) |

| Reagent or resource | Source | Identifier |
|---|---|---|
| Software and algorithms | | |
| ImageJ/Fiji | https://imagej.nih.gov/ij/ | |
| GraphPad Prism 9 software | https://www.graphpad.com/ | |
| Flowjo_v10.8 software | https://www.flowjo.com/ | |
| Image Studio Lite Version 5.2 | https://www.licor.com/bio/image-studio-lite/ | |
| Serial cloner 2.6.1 | https://serial-cloner.en.softonic.com/ | |
| SnapGene software | https://www.snapgene.com/ | |
| Synthego Performance Analysis, ICE Analysis (2019.v3.0) | https://www.synthego.com/ | |
| Galaxy Europe platform | (Afgan et al, 2018; Galaxy Community et al, 2022) https://usegalaxy.eu/ | |
| *Cutadapt* | https://cutadapt.readthedocs.io/en/stable/ | |
| *RNA STAR* | (Dobin et al, 2013) https://github.com/alexdobin/STAR | |
| *FeatureCounts* | (Dobin et al, 2013) https://bioconductor.org/packages/release/bioc/html/Rsubread.html | |
| DESeq2 | https://bioconductor.org/packages/release/bioc/html/DESeq2.html | |
| gProfiler platform | https://biit.cs.ut.ee/gprofiler/gost | |
| Venn diagrams | https://bioinformatics.psb.ugent.be/webtools/Venn/ | |

# Data Availability

Data from RNA-sequencing study are available at GEO database with the accession number GSE253680 (Fig 4).

# Supplementary Information

# Acknowledgements

The authors would like to acknowledge the core facilities of the Faculty of Sciences of the University of Lisbon (FCUL) and Centre for Ecology, Evolution and Environmental Changes (cE3c) & CHANGE. SG Martins was supported by Fundação para a Ciência e Tecnologia (FCT, Portugal; 2022.10813.BD) and the COST (European Cooperation in Science and Technology) Action CA20121 (Bench to bedside transition for pharmacological regulation of NRF2 in noncommunicable diseases, BenBedPhar) (E-COST-GRANT-CA20121-a076558c); AR Carlos by FCT (CEECIND/01589/2017) and L'Oréal Portugal Medals of Honor for Women in Science 2019; and S Thorsteinsdóttir by Association Française contre les Myopathies (AFM) Téléthon (contract no. 23049). SG Martins, V Ribeiro, C Melo, C Paulino-Cavaco, I Fonseca, B Saget, M Pita, DR Fernandes, P Gameiro dos Santos, G Rodrigues, R Zilhão, AR Carlos, and S Thorsteinsdóttir were supported by the donor Henrique Meirelles who chose to support the MATRIHEALTH Project (CC1036), FCT Project (10.54499/PTDC/BTM-ORG/1383/2020), and Unit Funding (10.54499/UIDB/00329/2020). F Herrera and F Murtinheira were supported by center grants to BioISI (10.54499/UIDB/04046/2020) and individual grants (10.54499/PTDC/FIS-MAC/2741/2021 and SFRH/BD/133220/2017, respectively) through FCT. Immunofluorescence experiments were supported by the Microscopy Facility of FCUL (PPBI-POCI-01-0145-FEDER-022122). Mouse breeding and maintenance were supported by CONGENTO LISBOA-01-0145-FEDER-022170 (FCT, Lisboa2020, Por2020, ERDF), and heterozygous crossing by FCUL animal house. Cell sorting was performed in the Flow Cytometry Facility of *Instituto de Medicina Molecular João Lobo Antunes* (IMM, Portugal). Flow cytometry analysis was performed in the Margarida Gama-Carvalho Laboratory at FCUL. RNA sequencing was performed in the Genomics Unit of *Instituto Gulbenkian de Ciência* (IGC, Portugal). The authors would also like to acknowledge the help of Dr. Edgar Gomes and Dr. Ana Raquel Pereira (Instituto de Medicina Molecular João Lobo Antunes, Faculdade de Medicina, Universidade de Lisboa, Portugal) with the protocol and materials for MuSC isolation.

## Author Contributions

SG Martins: conceptualization, data curation, formal analysis, validation, investigation, methodology, and writing—original draft, review, and editing.

V Ribeiro: data curation, formal analysis, investigation, methodology, and writing—original draft, review, and editing.

C Melo: data curation, formal analysis, investigation, and writing—review and editing.

C Paulino-Cavaco: data curation, formal analysis, investigation, methodology, and writing—review and editing.

D Antonini: data curation, software, formal analysis, and writing—review and editing.

S Dayalan Naidu: data curation, formal analysis, investigation, methodology, and writing—review and editing.

F Murtinheira: data curation, formal analysis, investigation, methodology, and writing—review and editing.

I Fonseca: data curation, formal analysis, investigation, methodology, and writing—review and editing.

B Saget: investigation, methodology, and writing—review and editing.

M Pita: data curation, methodology, and writing—review and editing.

DR Fernandes: data curation, software, formal analysis, investigation, methodology, and writing—original draft, review, and editing.

P Gameiro dos Santos: investigation, methodology, and writing—review and editing.

G Rodrigues: supervision and writing—review and editing.

R Zilhão: formal analysis, supervision, funding acquisition, project administration, and writing—review and editing.

F Herrera: supervision, funding acquisition, project administration, and writing—review and editing.

AT Dinkova-Kostova: supervision, funding acquisition, project administration, and writing—review and editing.

AR Carlos: conceptualization, formal analysis, supervision, funding acquisition, investigation, project administration, and writing—original draft, review, and editing.

S Thorsteinsdóttir: conceptualization, supervision, funding acquisition, project administration, and writing—original draft, review, and editing.

## Conflict of Interest Statement

The authors declare that they have no conflict of interest.

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
