## [Reviewer comments · Life Science Alliance]

Life Science Alliance

Laminin- α 2 chain deficiency in skeletal muscle causes dysregulation of multiple cellular mechanisms

Susana Martins, Vanessa Ribeiro, Catarina Melo, Cláudia Paulino-Cavaco, Dario Antonini, Sharadha Dayalan Naidu, Fernanda Murtinheira, Inês Fonseca, Bérénice Saget, Mafalda Pita, Diogo Fernandes, Pedro Gameiro dos Santos, Gabriela Rodrigues, Rita Zilhão, Federico Herrera, Albena Dinkova-Kostova, Ana Rita Carlos, and Sólveig Thorsteinsdóttir

DOI: <https://doi.org/10.26508/lsa.202402829>

Corresponding author(s): Ana Rita Carlos, University of Lisbon and Sólveig Thorsteinsdóttir, universidade de lisboa

Review Timeline:

Submission Date:	2024-05-17
Editorial Decision:	2024-06-14
Revision Received:	2024-09-11
Editorial Decision:	2024-09-12
Revision Received:	2024-09-13
Accepted:	2024-09-13

Transaction Report:

June 14, 2024

Re: Life Science Alliance manuscript #LSA-2024-02829-T

Prof. Ana Rita Carlos
Faculty of Sciences, University of Lisbon
Animal Biology Department - cE3c
Edifício C2, Piso 3, Gabinete 47A
Campo Grande
Lisbon 1749-016
Portugal

Dear Dr. Carlos,

Thank you for submitting your manuscript entitled "Laminin- α 2 chain deficiency in skeletal muscle causes dysregulation of multiple cellular mechanisms" to Life Science Alliance. The manuscript was assessed by expert reviewers, whose comments are appended to this letter. We invite you to submit a revised manuscript addressing the Reviewer comments.

Thank you for this interesting contribution to Life Science Alliance. We are looking forward to receiving your revised manuscript.

Sincerely,

B. MANUSCRIPT ORGANIZATION AND FORMATTING:

Reviewer #1 (Comments to the Authors (Required)):

The manuscript from SG Martins and colleagues reports how loss of Lama2 chain in muscle cells leads to dysregulation of multiple mechanisms. The authors generated CRISPR mutant clones for Lama2 deficiency using the C2C12 cell line to conduct several experiments, which pointed deficiencies in differentiation, proliferation and metabolism in the mutant cell lines. RNA seq studies were also performed on fetal muscles from control and mutant mice, highlighting several gene categories affected. While the manuscript describes a number of assays conducted with the mutant cell lines and multiple lists of differentially expressed genes, it lacks an overall conclusion of what are the main deficiencies that contribute to the severity of the disease. Further, it is felt that some of the data included is not totally novel or unexpected, given that previous studies reported similar problems with cells derived from human patients. The analyses of muscles from fetal animals (global KO) is also viewed as an incremental discovery, since similar findings were also reported for adult animals. Additional points to consider that could improve the manuscript:

- 1) The C2C12 model might have intrinsic properties that are not shared by primary muscle cells and therefore the observed phenotype might not exactly recapitulate what occurs in primary cells.
- 2) It appears that a single C2C12 WT clone and 2 mutant clones were compared. This seems insufficient and it lacks rigor, as intrinsic variability within independent biological replicates cannot be assessed. In proof of this concern, the data shows variability between the two mutant clones analyzed, as they can behave differently and raise the baseline variability for statistical analyses.
- 3) The differentiation assays performed on Lama 2 proficient and deficient cells suffer from time differences in reaching confluency and therefore the start of differentiation. Because WT cells grow faster, they will reach confluency at an earlier time and begin differentiation even though a high serum medium is present. One suggestion to control for this problem is to immunostain the samples for MyoD and myogenin as differentiation progresses, to evaluate the percentage of nuclei positive for these markers in each condition.
- 4) The western blots in Figures 1 and 3 seem to draw conclusions from technical replicates of the same clone(s), this seems insufficient to establish a robust conclusion. Multiple independent biological replicates should be compared.
- 5) Figure 6 and Figure 5 could be combined, since most of Figure 6 shows normalized plots of individual genes from the RNA seq data
- 6) The DEG should be combined in a single file, as one gene can be listed in multiple GO categories, making the data a bit redundant.
- 7) Autophagy flux should be measured in the presence or absence of autophagy blocking compound, which will be able to assess the accumulation of lipidated LC3 in the cells. One measurement is not sufficient to establish if autophagy is increased or decreased.

Reviewer #2 (Comments to the Authors (Required)):

The authors correctly highlight the lack of in-depth knowledge about the mechanisms that malfunction during the onset of LAMA2-CMD in utero. Their paper provides unique insights into the role of the Laminin- α 2 chain in muscle development and muscle cell homeostasis. The data are convincing, and this work may be critical in the development and study of future therapeutic approaches.

To strengthen the paper, there are some points that need clarification.

-The authors demonstrated that their KO cells exhibit proliferation and cell cycle defects. To analyze whether differentiation is impaired, the authors chose a cell-cell contact differentiation approach, but they did not indicate which marker was used to define T0. Typically, in standard differentiation protocols, T0 is when the differentiation media is added. This information is crucial to confirm that the delay in differentiation is due to the absence of Laminin- α 2 and not merely a result of delayed proliferation and, consequently, a delay in the initiation of cell-cell contact differentiation.

-The authors showed a lack of fusion in their KO cells. Did they check if the absence of Laminin- α 2 causes a downregulation of Myomixer and Myomaker? If yes could be important to show this data in the paper.

Reviewer #3 (Comments to the Authors (Required)):

1. Short summary, including description of the advance offered to the field.

The work presented by Martins et al., focuses on the description of yet not fully characterized molecular/cellular mechanisms caused by laminin-alpha2 chain deficiency in skeletal muscle, as defects or absence of laminin-alpha2 result in a rare but very severe form of congenital muscular dystrophy known as LAMA2-CMD. To study the consequences of laminin-alpha2 deficiency in murine skeletal muscle the authors relied on an immortalized murine myoblast cell line (C2C12) in which they deleted the Lama2 gene by means of the CRISPR/Cas9 editing tool. They demonstrated that laminin-alpha2 deficiency in vitro affects cells proliferation and myogenic differentiation program, boosts oxidative stress and decreases mitochondrial number thus impacting energy metabolism, and results in DNA damage and autophagy. These mechanisms were partially explored and characterized in previous studies on patient-derived myoblasts/myotubes and muscle samples isolated from adult LAMA2-CMD mice completely lacking laminin-alpha2 (as studied by De Oliveira et al., 2014 and Fontes-Oliveira et al., 2017, for example). However, the novelty of the proposed work relies on the more extensive characterization of these events in a cell line not previously exposed to the inflammatory/fibrotic milieu that characterises the dystrophic muscle (as Lama2 gene was deleted in healthy C2C12 myoblasts) and the confirmation of these findings in foetal LAMA2-CMD murine muscles, at developmental timepoints coinciding with the first pathological signs. The authors in fact performed RNA-sequencing analysis of foetal muscles (E17.5) isolated from the DyW knock-out LAMA2-CMD mouse model identifying a strong downregulation of genes affecting muscle development and growth and, comparing this transcriptomic signature with data obtained from available Chip-Seq database of C2C12 myoblasts and myocytes, they could identify pathways uniquely altered in laminin-alpha2 deficient muscles. Interestingly, most pathways altered in isolated foetal muscles at E17.5 reflected what observed in vitro. Finally, key events found to be altered both in vitro and at E17.5 were then studied also in muscle samples isolated from E18.5 DyW knock-out mice, to pinpoint the exact timing during which gene expression started to be altered. LAMA2-CMD onset was in fact previously reported to occur during these two timepoints by these authors (Nunes et al., 2017)

2. For each main point of the paper, please indicate if the data are strongly supportive. If not, explicitly state the additional experiments essential to support the claims made and the timeframe that these would require.

- Generation of Lama2-deficient C2C12 was obtained via CRISPR/Cas9 genome editing. PCR was used to confirm gene deletion combined with Sanger sequencing of the amplicon. Multiple (n=3) Lama2-knockout cell clones were studied to exclude that resulting phenotype was caused by off-target effects occurring in edited cells. Resazurin assay was used to monitor cell proliferation in wild-type and Lama2-deficient C2C12. Data presented in Figure1C clearly show reduced proliferation in knockout samples. From Figure1B onward, Lama2-knockout clones represented in figures are n=2, not clear if the two clones with the highest editing efficiency from figure 1B were chosen/which of the three clones represented in Figure1B were carried forward. Supplementary figure 1A - could the authors comment on why the deletion band was shown only in 1 KO sample? This is unclear. Block in G1 phase in KO cells demonstrated via propidium iodide staining followed by flow cytometry is convincing.

- Metabolic assays are convincing: measurement of reduced glutathione levels showed impairment of antioxidant activity in KO cells. Generally, there is high variability within KO samples and OH-1 Western Blot not clear in confirming increased OH-1 activity in both knockout samples. The authors should show more lanes of WT and KO to confirm the claimed difference. These data are complemented by Western Blot analysis of mitochondrial complexes. Fractionation analysis confirmed reduction of mitochondrial number in KO cells. Western Blot analysis of mitochondrial complexes is not entirely convincing as bands dimension in the knockout groups are not consistent. The authors should perform a functional metabolic assay on knockout and wild-type cells to confirm these data, and analyse mitochondria dimension (i.e. if they are swollen, enlarged etc) other than their number, given that the authors hypothesize non-functioning mitochondria. Measurement of 53BP1 foci used to monitor DNA double-strand breaks, indicative of DNA damage. However, monitoring the level of phosphorylated histone H2A would have been a more appropriate assay to investigate double-strand breaks.

- Apoptosis and autophagosome formation were investigated by qPCR analysis and Western blot analysis, respectively. Altered autophagic flux should be assessed via functional studies following the guidelines depicted in doi: 10.1080/15548627.2015.1100356.I

- Differentiation potential was monitored both at 5 and 14 days post differentiation. Myotubes were identified by myosin heavy chain staining as this protein is present only in terminally differentiated cells. Data presented in Figure 2 and 3 are robust in demonstrating differentiation defects via immunofluorescence, qPCR and Western blot analysis done at specific time-points.

- RNA-sequencing data analysis confirmed strong downregulation of developmental pathways at foetal stage. The parameters chosen for data analysis are consistent with what routinely used in the literature. Figure 5 panels show high homogeneity within DyW sample replicates only for downregulated genes, but the same trend is not observed in wild-type replicates or upregulated genes, could the authors comment on the reason behind this or the reference dataset used to generate this figure?

- The comparison of transcriptomic data with previously published Chip-seq database was useful to refine altered pathways. However, could the author briefly comment on why specific genes were chosen? For example, Tmem182 and Cacna1S chosen as top hits because of their role in the literature? In the panel in Figure4A these genes are not the most dysregulated.
 - Generally, legends fully explicative and inclusive of details regarding the numbers of experimental replicates.
 - Materials and methods well described and referenced.
 - The authors could consider to confirm at least a few findings obtained in C2C2 Lama2 KO cells in primary cells instead, obtainable directly from the DyW animal model, to be more consistent.
3. Lastly, indicate any additional issues you feel should be addressed
Minor comments are presented below:

Line 152: as results not significant, possibly rephrase with "SLC7A11 shows a trend towards reduction".

Line 210: by day 12? (not 14, according to the figure)

Line 194: "reduction was also evident for LC3-I"; no figure or data demonstrating this, could the author add evidence about it in the supplementary file?

Line 609: KO clones chosen are two (not three as indicated in Figure3A legend)

Line 992 (Reference 24): wrong reference info. Author is Burattini S et al., and year of publication is 2004.

Line 53-54 of Supplementary Figure legend: specify that analysis done in muscles isolated from E18.5 muscles.

Figure1J: while the panel showing quantification of 53BP1 clearly demonstrates an increase in the number of cells with more than 5 53BP1 foci, the immunofluorescence panels comparing WT and KO cells are not so clear in showing this. Maybe consider showing a different picture?

Figure 3: for consistency the authors could add a bar with n.s. stated for non-significant results, as done in previous figures.

Figure 6: not clear why in some cases used T-test and in others F-test

Could the authors comment on why autophagy markers were not considered in the study of foetal skeletal muscle samples

Supplementary Figure 4: for consistency the authors could add a bar with n.s. stated for non-significant results, as done in previous figures.

Laminin- α 2 chain deficiency in skeletal muscle causes dysregulation of multiple cellular mechanisms

Susana G Martins^{1,2}, Vanessa Ribeiro^{1,2}, Catarina Melo^{1,2}, Cláudia Paulino-Cavaco^{1,2}, Dario Antonini³, Sharadha Dayalan Naidu⁴, Fernanda Murtinheira^{5,6}, Inês Fonseca^{1,2}, Bérénice Saget^{1,2}, Mafalda Pita^{1,2}, Diogo R Fernandes^{1,2}, Pedro Gameiro dos Santos^{1,2}, Gabriela Rodrigues^{1,2}, Rita Zilhão^{1,7}, Federico Herrera^{5,6}, Alben T Dinkova-Kostova⁴, Ana Rita Carlos^{1,2,*,\$} & Sólveig Thorsteinsdóttir^{1,2,*,\$}.

¹ Centre for Ecology, Evolution and Environmental Changes (cE3c) & CHANGE, Faculdade de Ciências, Universidade de Lisboa, 1749-016 Lisboa, Portugal

² Departamento de Biologia Animal, Faculdade de Ciências, Universidade de Lisboa, 1749-016 Lisboa, Portugal

³ Department of Biology, University of Naples "Federico II", 80126 Naples, Italy

⁴ Jacqui Wood Cancer Centre, Division of Cellular and Systems Medicine, School of Medicine, University of Dundee, Dundee DD1 9SY, Scotland, U.K.

⁵ Biosystems & Integrative Sciences Institute, Faculdade de Ciências, Universidade de Lisboa, 1649-004 Lisboa, Portugal

⁶ Departamento de Química e Bioquímica, Faculdade de Ciências, Universidade de Lisboa, 1749-016 Lisboa, Portugal

⁷ Departamento de Biologia Vegetal, Faculdade de Ciências, Universidade de Lisboa, 1749-016 Lisboa, Portugal

*co-senior authors.

\$Corresponding authors: arcarlos@ciencias.ulisboa.pt, solveig@ciencias.ulisboa.pt

Reviewer #1 (Comments to the Authors (Required)):

Comment #1: *The manuscript from SG Martins and colleagues reports how loss of Lama2 chain in muscle cells leads to dysregulation of multiple mechanisms. The authors generated CRISPR mutant clones for Lama2 deficiency using the C2C12 cell line to conduct several experiments, which pointed deficiencies in differentiation, proliferation and metabolism in the mutant cell lines. RNA seq studies were also performed on fetal muscles from control and mutant mice, highlighting several gene categories affected. While the manuscript describes a number of assays conducted with the mutant cell lines and multiple lists of differentially expressed genes, it lacks an overall conclusion of what are the main deficiencies that contribute to the severity of the disease. Further, it is felt that some of the data included is not totally novel or unexpected, given that previous studies reported similar problems with cells derived from human patients. The analyses of muscles from fetal animals (global KO) is also viewed as an incremental discovery, since similar findings were also reported for adult animals. Additional points to consider that could improve the manuscript:*

Reply to #1: We thank the reviewer for the overall comments on our manuscript. Even though some of the data shown here have been explored in previous studies, none of the studies clarified which were the key mechanism

found to be altered at disease onset. Our results provide unique insights into the onset of the disease, characterized by a marked downregulation of gene expression. This downregulation impacts vital processes such as cytoskeletal organization, myoblast differentiation and fusion, DNA repair, and responses to oxidative stress. In particular, we observe a profound decrease in the expression of specific myosin heavy chain genes and calcium regulators, suggesting a specific link between LAMA2 and these cellular components. We have edited the manuscript, particularly the abstract and the discussion, to make our conclusions more clear. This involved: (1) reviewing the nomenclature of muscle cells (i.e. myoblast, myofiber /muscle fiber) to make sure it is always evident which cell type we are referring to, and using the term “muscle differentiation” as a general term covering the whole process from myoblast to muscle fibers); (2) shortening and/or restructuring sentences to make the message more clear; (3) adding two new sections, one in the abstract and the other one at the end of the discussion, as follows:

Abstract: “We generated Lama2-deficient C2C12 cells and found that Lama2-deficient myoblasts display proliferation, differentiation and fusion defects, DNA damage, oxidative stress and mitochondrial dysfunction. Moreover, fetal myoblasts isolated from the dy^W mouse model of LAMA2-CMD display impaired differentiation and fusion in vitro. We also showed that disease onset during fetal development is characterized by a significant downregulation of gene expression in muscle fibers, causing pronounced effects on cytoskeletal organization, muscle differentiation, as well as altered DNA repair and oxidative stress responses.”

Discussion: “Specifically, we have shown that Lama2-deficient myoblasts cultured in vitro increase NFIX levels and fail to upregulate Myogenin and Myomaker, leading to an inability to fuse. Moreover, we have shown that the expression of key genes associated with muscle differentiation, including Myh2, Myh7, Tmem182 and Cacna1s, is severely impaired in the absence of Lama2, as soon as laminin- α 2 assumes its role as the central laminin isoform in the basement membrane of muscle fibers.”

Comment #2: *1) The C2C12 model might have intrinsic properties that are not shared by primary muscle cells and therefore the observed phenotype might not exactly recapitulate what occurs in primary cells.*

Reply to #2: To expand and further support our findings and to address the comments of Reviewer 1 (comment #2) and Reviewer 3 (comment #21), we isolated primary mouse myoblast from E16.5-19.5 fetuses. The myoblasts isolated from each fetus were kept individually and two experimental approaches were performed: analysis of number of myofibers (based on myosin heavy chain staining) and mRNA expression of two key differentiation markers shown to be downregulated in dy^W muscle fibers and Lama2-deficient C2C12 myotubes. The following sentences and figures were added to the results section:

“To further extend this analysis, we isolated primary fetal mouse myoblasts from wildtype and dy^W fetuses and found that dy^W myoblasts display a differentiation defect, which is evident by the significant reduction in the number

of myofibers formed when compared to wildtype, already three days after being placed in differentiation medium (Figure 2C)."

"The expression of both Tmem182 and Cacna1s was also found to be reduced in dyW primary fetal myoblasts, when compared to wildtype myoblast, three days after the addition of differentiation medium (Figures 5L,M)."

Comment #3: *2) It appears that a single C2C12 WT clone and 2 mutant clones were compared. This seems insufficient and it lacks rigor, as intrinsic variability within independent biological replicates cannot be assessed. In proof of this concern, the data shows variability between the two mutant clones analyzed, as they can behave differently and raise the baseline variability for statistical analyses.*

Reply to #3: To clarify this point, all the experiments were performed with WT C2C12 cells (not a single cell clone) and with at least two C2C12 *Lama2* KO single cell clones. In this work we used 4 independent C2C12 *Lama2* KO single cell clones, all displaying similar phenotypes. The reason we used different clones throughout the project is that C2C12 *Lama2* KO cells display characteristics (e.g. reduced proliferation, increased DNA damage), and thus cannot be maintained in culture for many passages. Each experiment was performed at least three times, and WT and *Lama2* KO cells were collected each time, corresponding to independent biological replicates.

To clarify this issue, the following sentence was added to the results section:

"Due to this defect, Lama2 knockout single cell clones were passaged for a limited time and new clones were periodically generated and validated as described above."

Comment #4: *3) The differentiation assays performed on Lama 2 proficient and deficient cells suffer from time differences in reaching confluency and therefore the start of differentiation. Because WT cells grow faster, they will reach confluency at an earlier time and begin differentiation even though a high serum medium is present. One suggestion to control for this problem is to immunostain the samples for MyoD and myogenin as differentiation progresses, to evaluate the percentage of nuclei positive for these markers in each condition.*

Reply to #4: We were aware of this problem and to overcome the proliferation defect we plated WT and KO cells at a high density, allowing them to reach confluency faster (i.e. before the proliferation rate became significantly different between them). Even though MyoD and myogenin are key markers for muscle differentiation, in Figure 3 we show that some myogenic regulatory factors, including myogenin, and their target genes are altered in the *Lama2* KO cells, therefore some of the conclusions drawn from the comparison of myogenin between WT and KO during the differentiation process might be inconclusive.

To clarify our differentiation protocol, we have added more detailed information in the following sections:

Results:

"In addition, cells were plated at high density to allow confluency to be reached within the first 48h (Supplementary Figure 3A), prior to significant differences in proliferation."

Materials and methods:

“Wildtype and Lama2 knockout C2C12 cells were plated (Day -1) in complete DMEM medium at a density of 50 000 cells per well in a 24-well plate with coverslips and left to differentiate until day 5 or 14.”

Figure legend:

“Figure 2 Lama2 knockout cells show impaired differentiation. A) C2C12 wildtype (WT) and two independent Lama2 knockout single cell clones (KO) were plated (Day -1), cultured until Day 5 to induce differentiation and then fixed and processed for immunofluorescence with an anti-myosin heavy chain antibody.”

Comment #5: *4) The western blots in Figures 1 and 3 seem to draw conclusions from technical replicates of the same clone(s), this seems insufficient to establish a robust conclusion. Multiple independent biological replicates should be compared.*

Reply to #5: As clarified in the reply to Comment #3, each experiment was performed at least three times, and WT and Lama2 KO cells were collected each time, corresponding to independent biological replicates.

Comment #6: *5) Figure 6 and Figure 5 could be combined, since most of Figure 6 shows normalized plots of individual genes from the RNA seq data*

Reply to #6: Former Figures 4 and 5 were combined (current Figure 4) to place all the bioinformatic analysis in the same figure. The text and remaining figures have been adjusted accordingly. Former Figure 6 (currently Figure 5) includes the validation of the RNA seq data in the 3 models used: C2C12 cells, dy^{W} fetuses and primary fetal myoblasts.

Comment #7: *6) The DEG should be combined in a single file, as one gene can be listed in multiple GO categories, making the data a bit redundant.*

Reply to #7: The tables provided as supplementary information focus on the GO analysis. We agree that some information is redundant, since the same gene can be found in different GO categories. However, we consider that highlighting the genes from these key GO is relevant for our study and for the readers, providing a unique list of potential target genes that can be further explored in future studies.

Comment #8: *7) Autophagy flux should be measured in the presence or absence of autophagy blocking compound, which will be able to assess the accumulation of lipidated LC3 in the cells. One measurement is not sufficient to establish if autophagy is increased or decreased.*

Reply to #8: In order to complete our analysis for autophagy and address the concerns from both Reviewer 1 (comment #8) and Reviewer 3 (comment #18), we analysed the expression of two essential autophagy markers, *Atg5* and *Atg7*, in C2C12 Lama2-proficient and deficient cells. The following sentence and corresponding figure were added to the results section:

“Additionally, we analyzed the expression of two essential components of the conventional autophagy pathway, ATG5 and ATG7 (Arakawa et al, 2017), and found that Atg7 was downregulated in Lama2-deficient C2C12 cells when compared to their wildtype counterparts (Supplementary Figure 2E).”

Additionally, considering the important role of autophagy in myoblast differentiation, we investigated the expression of *Atg5* and *Atg7* in cells 5 days post differentiation. The following sentences and corresponding figure were added to the results section:

“Since we have shown that autophagy is compromised in Lama2-deficient C2C12 myoblasts (Supplementary Figures 2D,E) and that autophagy is known to be important for myoblast differentiation (McMillan & Quadriatero, 2014), we investigated whether the expression of Atg5 and Atg7 was impaired in Lama2-deficient cells at day 5 post-differentiation (Figure 3G). We found that the expression of both Atg5 and Atg7 is significantly downregulated in Lama2-deficient cells in comparison to the wildtype.”

Reviewer #2 (Comments to the Authors (Required)):

Comment #9: *The authors correctly highlight the lack of in-depth knowledge about the mechanisms that malfunction during the onset of LAMA2-CMD in utero. Their paper provides unique insights into the role of the Laminin- α 2 chain in muscle development and muscle cell homeostasis. The data are convincing, and this work may be critical in the development and study of future therapeutic approaches.*

Reply to #9: We thank the reviewer for this comment, which coincides with our vision of the impact of the study

To strengthen the paper, there are some points that need clarification.

Comment #10: *-The authors demonstrated that their KO cells exhibit proliferation and cell cycle defects. To analyze whether differentiation is impaired, the authors chose a cell-cell contact differentiation approach, but they did not indicate which marker was used to define T0. Typically, in standard differentiation protocols, T0 is when the differentiation media is added. This information is crucial to confirm that the delay in differentiation is due to the absence of Laminin- α 2 and not merely a result of delayed proliferation and, consequently, a delay in the initiation of cell-cell contact differentiation.*

Reply to #10: To clarify this point, we have added more detailed information in the following sections:

Results:

“In addition, cells were plated at high density to allow confluency to be reached within the first 48h (Supplementary Figure 3A), prior to significant differences in proliferation.”

Materials and methods:

“Wildtype and Lama2 knockout C2C12 cells were plated (Day -1) in complete DMEM medium at a density of 50 000 cells per well in a 24-well plate with coverslips and left to differentiate until day 5 or 14.”

Figure legend:

“Figure 2 Lama2 knockout cells show impaired differentiation. A) C2C12 wildtype (WT) and two independent Lama2 knockout single cell clones (KO) were plated (Day -1), cultured until Day 5 to induce differentiation and then fixed and processed for immunofluorescence with an anti-myosin heavy chain antibody.”

Comment #11: *-The authors showed a lack of fusion in their KO cells. Did they*

check if the absence of Laminin- α 2 causes a downregulation of Myomixer and Myomaker? If yes could be important to show this data in the paper.

Reply to #11: To address this point we analyzed the expression of *Mymk* (encoding Myomaker) and *Mymx* (encoding Myomerger/Myomixer) in *Lama2*-deficient and wildtype C2C12 myoblasts and added the following sentence to the results section, as well as the corresponding figures:

“To understand if the differentiation defect could also be linked with an impairment in myoblast fusion, we analyzed the expression of two key players in myoblast fusion, Myomaker and Myomerger (also known as Myomixer or Minion), which act in a sequential manner to promote plasma membrane remodeling and fusion (Leikina et al, 2018). Myomaker acts first, promoting hemifusion of the adjacent plasma membranes, while Myomerger completes the fusion process (Leikina et al, 2018). Our results showed that Lama2-deficient cells have impaired Mymk expression (encoding Myomaker) (Figure 3H), indicating that fusion is compromised already at the hemifusion stage. Additionally, there is a trend towards reduced expression of Mymx (encoding Myomerger) (Figure 3I). This suggests that Lama2-deficient C2C12 myoblasts lack the ability to fuse.”

Reviewer #3 (Comments to the Authors (Required)):

Comment #12: *1. Short summary, including description of the advance offered to the field. The work presented by Martins et al., focuses on the description of yet not fully characterized molecular/cellular mechanisms caused by laminin-alpha2 chain deficiency in skeletal muscle, as defects or absence of laminin-alpha2 result in a rare but very severe form of congenital muscular dystrophy known as LAMA2-CMD. To study the consequences of laminin-alpha2 deficiency in murine skeletal muscle the authors relied on an immortalized murine myoblast cell line (C2C12) in which they deleted the Lama2 gene by means of the CRISPR/Cas9 editing tool. They demonstrated that laminin-alpha2 deficiency in vitro affects cells proliferation and myogenic differentiation program, boosts oxidative stress and decreases mitochondrial number thus impacting energy metabolism, and results in DNA damage and autophagy. These mechanisms were partially explored and characterized in previous studies on patient-derived myoblasts/myotubes and muscle samples isolated from adult LAMA2-CMD mice completely lacking laminin-alpha2 (as studied by De Oliveira et al., 2014 and Fontes-Oliveira et al., 2017, for example). However, the novelty of the proposed work relies on the more extensive characterization of these events in a cell line not previously exposed to the inflammatory/fibrotic milieu that characterises the dystrophic muscle (as Lama2 gene was deleted in healthy C2C12 myoblasts) and the confirmation of these findings in foetal LAMA2-CMD murine muscles, at developmental timepoints coinciding with the first pathological signs. The authors in fact performed RNA-sequencing analysis of foetal muscles (E17.5) isolated from the DyW knock-out LAMA2-CMD mouse model identifying a strong downregulation of genes affecting muscle development and growth and, comparing this transcriptomic signature with data obtained from available Chip-Seq database of C2C12 myoblasts and myocytes, they could identify pathways uniquely altered in laminin-alpha2 deficient muscles. Interestingly, most pathways altered in isolated foetal muscles at E17.5 reflected what observed in vitro. Finally, key events found to be altered both in vitro and at E17.5 were then studied also in muscle samples isolated*

from E18.5 DyW knock-out mice, to pinpoint the exact timing during which gene expression started to be altered. LAMA2-CMD onset was in fact previously reported to occur during these two timepoints by these authors (Nunes et al., 2017)

Reply to #12: We thank the reviewer for the thorough analysis of our manuscript, highlighting the most significant advances from our study.

2. For each main point of the paper, please indicate if the data are strongly supportive. If not, explicitly state the additional experiments essential to support the claims made and the timeframe that these would require.
- Generation of Lama2-deficient C2C12 was obtained via CRISPR/Cas9 genome editing. PCR was used to confirm gene deletion combined with Sanger sequencing of the amplicon. Multiple (n=3) Lama2-knockout cell clones were studied to exclude that resulting phenotype was caused by off-target effects occurring in edited cells. Resazurin assay was used to monitor cell proliferation in wild-type and Lama2-deficient C2C12. Data presented in Figure1C clearly show reduced proliferation in knockout samples.

Comment #13: *From Figure1B onward, Lama2-knockout clones represented in figures are n=2, not clear if the two clones with the highest editing efficiency from figure 1B were chosen/which of the three clones represented in Figure1B were carried forward.*

Reply to #13: All the experiments were performed with at least two independent single cell clones. In this work we used 4 independent C2C12 *Lama2* KO single cell clones, all displaying similar phenotypes. The reason we used different clones throughout the project is that C2C12 *Lama2* KO cells display characteristics (e.g. reduced proliferation, increased DNA damage), and thus cannot be maintained in culture for many passages. To clarify this point, the following sentence was added to the Results sections:

“Due to this defect, Lama2 knockout single cell clones were passaged for a limited time and new clones were periodically generated and validated as described above.”

Comment #14: *Supplementary figure 1A - could the authors comment on why the deletion band was shown only in 1 KO sample? This is unclear.*

Reply to #14: From the *Lama2* KO single cell clones that we have sequenced, most displayed short nucleotide deletions in the two exons of *Lama2* that were targeted by CRISPR-Cas9. Using PCR and agarose gels does not provide enough resolution to identify such small changes as bands with different molecular weights. In contrast, large deletions, as observed for one of the alleles in one of the single cell clones can be detected as a band with a different molecular weight.

Block in G1 phase in KO cells demonstrated via propidium iodide staining followed by flow cytometry is convincing.

Comment #15: *- Metabolic assays are convincing: measurement of reduced glutathione levels showed impairment of antioxidant activity in KO cells. Generally, there is high variability within KO samples and OH-1 Western Blot not clear in confirming increased OH-1 activity in both knockout samples. The*

authors should show more lanes of WT and KO to confirm the claimed difference. These data are complemented by Western Blot analysis of mitochondrial complexes.

Reply to #15: We have replaced the representative western blot image with another image where the staining of the two KO lanes is more similar and therefore reflects the quantification shown.

Comment #16: *Fractionation analysis confirmed reduction of mitochondrial number in KO cells. Western Blot analysis of mitochondrial complexes is not entirely convincing as bands dimension in the knockout groups are not consistent. The authors should perform a functional metabolic assay on knockout and wild-type cells to confirm these data, and analyse mitochondria dimension (i.e. if they are swollen, enlarged etc) other than their number, given that the authors hypothesize non-functioning mitochondria.*

Reply to #16: To further characterize changes in mitochondria, we undertook an analysis of the morphology of mitochondria and added the following text and corresponding figures to the results:

“Notably, an increase in mitochondrial branching (Leduc-Gaudet et al, 2015) and a decrease in mitochondrial number (Crane et al, 2010) have also been associated with skeletal muscle aging. Therefore, we next analyzed the percentage of branched mitochondria (Figures 1I-K) and the mitochondrial number (Figure 1L) using Lama2-proficient vs. deficient cells. We found a significant increase in both the number of mitochondrial branches and their total length in Lama2-deficient cells when compared to wildtype cells (Figures 1J,K). This suggests that in the absence of Lama2, mitochondria decrease their fission and increase their fusion levels, as previously shown to occur in aged muscle (Leduc-Gaudet et al, 2015).”

Comment #17: *Measurement of 53BP1 foci used to monitor DNA double-strand breaks, indicative of DNA damage. However, monitoring the level of phosphorylated histone H2A would have been a more appropriate assay to investigate double-strand breaks.*

Reply to #17: Both 53BP1 and phosphorylated H2AX are bona fide markers of DNA double strand breaks (PMID: 21868291, PMID: 24326623). The anti-53BP1 antibody used in our study has a good signal-to-noise ratio in immunofluorescence, for C2C12 cells, and was therefore used for this analysis.

Comment #18: *Apoptosis and autophagosome formation were investigated by qPCR analysis and Western blot analysis, respectively. Altered autophagic flux should be assessed via functional studies following the guidelines depicted in doi: 10.1080/15548627.2015.1100356.*

Reply to #18: In order to complete our analysis for autophagy and address the concerns from both Reviewer 1 (comment #8) and Reviewer 3 (comment #18), we analysed the expression of two essential autophagy markers, *Atg5* and *Atg7*, in C2C12 *Lama2*-proficient and deficient cells. The following sentence and corresponding figure were added to the results section:

“Additionally, we analyzed the expression of two essential components of the conventional autophagy pathway, ATG5 and ATG7 (Arakawa et al, 2017), and found that Atg7 was downregulated in Lama2-deficient C2C12 cells when compared to their wildtype counterparts (Supplementary Figure 2E).”

Additionally, considering the important role of autophagy in myoblast differentiation, we investigated the expression of Atg5 and Atg7 in cells 5 days post differentiation. The following sentences and corresponding figure were added to the results section:

“Since we have shown that autophagy is compromised in Lama2-deficient C2C12 myoblasts (Supplementary Figures 2D,E) and that autophagy is known to be important for myoblast differentiation (McMillan & Quadriatero, 2014), we investigated whether the expression of Atg5 and Atg7 was impaired in Lama2-deficient cells at day 5 post-differentiation (Figure 3G). We found that the expression of both Atg5 and Atg7 is significantly downregulated in Lama2-deficient cells in comparison to the wildtype.”

- Differentiation potential was monitored both at 5 and 14 days post differentiation. Myotubes were identified by myosin heavy chain staining as this protein is present only in terminally differentiated cells. Data presented in Figure 2 and 3 are robust in demonstrating differentiation defects via immunofluorescence, qPCR and Western blot analysis done at specific time-points.

Comment #19: *- RNA-sequencing data analysis confirmed strong downregulation of developmental pathways at foetal stage. The parameters chosen for data analysis are consistent with what routinely used in the literature. Figure 5 panels show high homogeneity within DyW sample replicates only for downregulated genes, but the same trend is not observed in wild-type replicates or upregulated genes, could the authors comment on the reason behind this or the reference dataset used to generate this figure?*

Reply to #19: The dataset used for this analysis was the DEG from the RNAseq analysis performed as described in Figure 4A-C, which were analyzed using Venn diagram analysis for different GO categories. For clarity the figure legend of Figure 4 is now more detailed:

*“Figure 4.... **F-H**) Heatmaps of 20 representative genes (15 downregulated and 5 upregulated) from the Venn diagram analysis comparing the differentially expressed genes (DEGs) (adjusted p-value 0.05, log2 fold change +/-1.5) of wildtype vs. dyW muscle fibers **from the RNA seq analysis in Figure 4A** with the following gene ontologies: (A) GO:0030036 Actin Cytoskeleton Organization and GO:0042692 Muscle Cell Differentiation. (B) GO:0007049 Cell Cycle and GO:0006281 DNA repair. (C) GO:0006979 Response to Oxidative Stress and GO:0007005 Mitochondria organization.”*

As for the variability found in this analysis, it is possible that absence of functional *Lama2* completely abolishes the expression of specific genes, and therefore there is a consistency between the different dyW samples, while WT fetuses display a range of expression levels, in accordance with the highly dynamic nature of embryonic development. One can speculate that absence of *Lama2* also triggers the upregulation of specific genes, but to a lesser extent and therefore the expression is variable. This is supported by the observation that there is a larger number of downregulated genes, compared to upregulated genes, in the dyW fetal muscle fibers.

Comment #20: *- The comparison of transcriptomic data with previously*

published Chip-seq database was useful to refine altered pathways. However, could the author briefly comment on why specific genes were chosen? For example, Tmem182 and Cacna1s chosen as top hits because of their role in the literature? In the panel in Figure 4A these genes are not the most dysregulated.

Reply to #20: Our RNAseq analysis of WT vs dyW fetal muscle fibers revealed a large number of downregulated genes, from a variety of pathways, which was not easy to interpret. Therefore, since our *in vitro* analysis revealed that a differentiation defect is a key aspect of *Lama2*-deficiency, we compared previously published ChIP-seq dataset of C2C12 Myoblasts and Myocytes (H3K4me3 peaks) from ENCODE Project (<https://www.encodeproject.org/help/citing-encode/>) with our RNAseq dataset. *Tmem182* and *Cacna1s* were some of the DEG that were identified as key target genes in our comparative analysis with the previously published ChIP-seq dataset. These genes have been shown to be important for muscle cell differentiation and cytoskeleton organization, as referred in the manuscript.

To clarify this point, we added the following information to the Results section:

“Cacna1s and Tmem182 were also identified as common genes in our comparison between DEGs from our muscle fibers RNAseq with C2C12 myoblast and C2C12 myocyte ChIP-seq analysis for H3K4me3 (Supplementary Table 1).”

In addition, both genes are now shown in bold in the Supplementary Table 1.

- Generally, legends fully explicative and inclusive of details regarding the numbers of experimental replicates.

Materials and methods well described and referenced.

Comment #21: *- The authors could consider to confirm at least a few findings obtained in C2C2 *Lama2* KO cells in primary cells instead, obtainable directly from the DyW animal model, to be more consistent.*

Reply to #21: To address the comments of Reviewer 1 (comment #2) and Reviewer 3 (comment #21), we isolated primary fetal mouse myoblast from E16.5-19.5 fetuses. The myoblasts isolated from each fetus were kept individually and two experimental approaches were performed: analysis of the number of myofibers formed (based on myosin heavy chain staining) and mRNA expression of two key differentiation markers shown to be downregulated in *dy^W* muscle fibers and *Lama2*-deficient C2C12 myotubes. The following sentences and figures were added to the results section:

“To further extend this analysis, we isolated primary fetal mouse myoblasts from wildtype and dyW fetuses and found that dyW myoblasts display a differentiation defect, which is evident by the significant reduction in the number of myofibers formed when compared to wildtype, already three days after being placed in differentiation medium (Figure 2C).”

*“The expression of both *Tmem182* and *Cacna1s* was also found to be reduced in dyW primary fetal myoblasts, when compared to wildtype myoblast, three days after the addition of differentiation medium (Figures 5L,M).”*

3. Lastly, indicate any additional issues you feel should be addressed
Minor comments are presented below:

Comment #22: Line 152: as results not significant, possibly rephrase with "SLC7A11 shows a trend towards reduction".

Reply to #22: This sentence has now been rephrased as follows:

"However, we found that the cystine/glutamate transporter SLC7A11 showed a trend towards reduction in Lama2-deficient cells (Figure 1F)."

Comment #23: Line 210: by day 12? (not 14, according to the figure)

Reply to #23: For accuracy, we have now rephrased as follows:

"Lama2-proficient cells formed evident myotube-like structures by day 5, while similar structures, even though rare, could only be observed from day 12 onwards in Lama2-deficient cells (Supplementary Figure 3A)."

Comment #24: Line 194: "reduction was also evident for LC3-I"; no figure or data demonstrating this, could the author add evidence about it in the supplementary file?

Reply to #24: We have rephrased as follows:

"This reduction was also apparent for LC3-I, but since it was often undetectable in Lama2-deficient cells, quantification was not possible."

Comment #25: Line 609: KO clones chosen are two (not three as indicated in Figure3A legend)

Reply to #25: This has now been corrected.

Comment #26: Line 992 (Reference 24): wrong reference info. Author is Burattini S et al., and year of publication is 2004.

Reply to #26: This reference has now been corrected:

"Burattini S, Ferri R, Battistelli M, Curci R, Luchetti F, Falcieri E (2004) C2C12 murine myoblasts as a model of skeletal muscle development: Morpho-functional characterization. European Journal of Histochemistry 48: 223–233. doi:10.4081/891."

Comment #27: Line 53-54 of Supplementary Figure legend: specify that analysis done in muscles isolated from E18.5 muscles.

Reply to #27: This has been corrected as follows:

"Densitometry analysis of protein extracts obtained from epaxial muscles collected from wildtype (WT) and dyW fetuses at E18.5. The extracts were analyzed by western blot and the membranes probed with OXPHOS (CII - complex II; CIII - complex III; CIV - complex IV; CV - complex V) antibody cocktail as shown in Figure 7C."

Comment #28: Figure1J: while the panel showing quantification of 53BP1 clearly demonstrates an increase in the number of cells with more than 5 53BP1 foci, the immunofluorescence panels comparing WT and KO cells are not so clear in showing this. Maybe consider showing a different picture?

Reply to #28: The immunofluorescence panels have been replaced.

Comment #29: Figure 3: for consistency the authors could add a bar with n.s. stated for non-significant results, as done in previous figures.

Reply to #29: This has now been corrected.

Comment #30: *Figure 6: not clear why in some cases used T-test and in others F-test*

Reply to #30: While the differences between WT and dyW samples shown in Figure 6 are obvious, statistical significance was not always reached using a T-test. The F-test serves to demonstrate this difference statistically.

Comment #31: *Could the authors comment on why autophagy markers were not considered in the study of foetal skeletal muscle samples*

Reply to #31: Taking into account the important role of calcium in several cell mechanisms, including autophagy (Bootman et al, 2018), and the key role of autophagy in myoblast differentiation (McMillan & Quadrilatero, 2014), we analyzed the expression of *Atg5* and *Atg7* in whole muscle of wildtype and *dy^W* fetuses (**Reviewers Figure 1**). We found that the expression of both genes was increased in *dy^W* fetal muscles at E18.5 when compared to WT fetuses. This increase was significant for *Atg7*. This may suggest an attempt to compensate for the differentiation defect *in vivo*.

Reviewers Figure 1 Expression of autophagy markers in whole muscle of wildtype and *dy^W* fetuses. Epaxial muscles were collected from wildtype (WT) and *dy^W* fetuses at E18.5 and RNA was extracted. The expression of the autophagy markers *Atg5* and *Atg7* was then analyzed by qPCR. Expression levels were normalized using the housekeeping gene *Rplp0*. n=4-7 fetuses per genotype. Statistical analysis was performed using Unpaired T-test. P-value: * p<0.05.

Comment #32: *Supplementary Figure 4: for consistency the authors could add a bar with n.s. stated for non-significant results, as done in previous figures.*

Reply to #32: This has now been corrected.

September 12, 2024

RE: Life Science Alliance Manuscript #LSA-2024-02829-TR

Prof. Ana Rita Carlos
University of Lisbon
Animal Biology Department - cE3c
Edifício C2, Piso 3, Gabinete 47A
Campo Grande
Lisbon 1749-016
Portugal

Dear Dr. Carlos,

Thank you for submitting your revised manuscript entitled "Laminin- α 2 chain deficiency in skeletal muscle causes dysregulation of multiple cellular mechanisms". We would be happy to publish your paper in Life Science Alliance pending final revisions necessary to meet our formatting guidelines.

- please be sure that the authorship listing and order is correct
- please consult our manuscript preparation guidelines <https://www.life-science-alliance.org/manuscript-prep> and make sure your manuscript sections are in the correct order
- please add your supplemental figure legends to the main figure legend section
- please deposit the RNA-seq dataset and include accession information in a Data Availability statement, placed at the end of the Materials and Methods section
- the Cover Art file should instead be uploaded as a Graphical Abstract

LSA now encourages authors to provide a 30-60 second video where the study is briefly explained. We will use these videos on social media to promote the published paper and the presenting author (for examples, see <https://docs.google.com/document/d/1-UWCfbE4pGcDdcgzcmiuJI2XMBJnxKYeqRvLLrLS08s/edit?usp=sharing>). Corresponding or first-authors are welcome to submit the video. Please submit only one video per manuscript. The video can be emailed to contact@life-science-alliance.org

A. FINAL FILES:

B. MANUSCRIPT ORGANIZATION AND FORMATTING:

Sincerely,

September 13, 2024

RE: Life Science Alliance Manuscript #LSA-2024-02829-TRR

Prof. Ana Rita Carlos
University of Lisbon
Animal Biology Department - cE3c
Edifício C2, Piso 3, Gabinete 47A
Campo Grande
Lisbon 1749-016
Portugal

Dear Dr. Carlos,

Thank you for submitting your Research Article entitled "Laminin- α 2 chain deficiency in skeletal muscle causes dysregulation of multiple cellular mechanisms". It is a pleasure to let you know that your manuscript is now accepted for publication in Life Science Alliance. Congratulations on this interesting work.

DISTRIBUTION OF MATERIALS:

Again, congratulations on a very nice paper. I hope you found the review process to be constructive and are pleased with how the manuscript was handled editorially. We look forward to future exciting submissions from your lab.

Sincerely,
